



# Drastic decline of floodpulse in the Cambodian floodplains (the Mekong River and the Tonle Sap system)

Samuel De Xun Chua[1], Xi Xi Lu[1], Chantha Oeurng[2], Ty Sok[2,3], Carl Grundy-Warr[1]

[1]Department of Geography, National University of Singapore, Singapore

[2]Faculty of Hydrology and Water Resources Engineering, Institute of Technology of Cambodia, Russian Federation Blvd., P.O. BOX 86, Phnom Penh, Cambodia.

[3]Laboratoire Ecologie fonctionnelle et environnement, Université de Toulouse, CNRS, Toulouse INP, Université Toulouse 3 – Paul Sabatier (UT3), Toulouse, France.

*Correspondence to*: Lu Xi Xi (geoluxx@nus.edu.sg)

**Abstract.** The Cambodian floodplains experience a yearly floodpulse that is essential to sustain fisheries and the agricultural calendar. Sixty years of data from 1960-2019 are used to track the changes to the floodpulse there. We find that minimum water levels in 2010-2019 have increased by up to 1.55m at Kratie and maximum water levels have decreased by up to 0.79m at Prek Kdam when compared to 1960-1991 levels, causing a reduction of the annual flood extent. Concurrently, the duration of the flooding season has decreased by about 26 days (Kompong Cham) – 40 days (Chaktomuk), with the season

starting later and ending much earlier. Along the Tonle Sap River, the average annual reverse flow from the Mekong to the Tonle Sap Lake has decreased by 56.5%, from 48.7 $km^3$ in 1962-1972 to 31.7 $km^3$ in 2010-2018. As a result, wet-season water levels at Tonle Sap Lake has dropped by 1.05m in 2010-2019 since 1996-2009, corresponding to a 20.6% shrinkage of the Lake area. In addition to known upstream contributors such as hydropower dams, two anthropogenic causes of the drastic alterations to the floodpulse are identified: irrigation and channel incision. We estimate that water withdrawal in the

Cambodian floodplains is occurring at a rate of $(2.1 \pm 0.3)$ $km^3$/yr and incision-induced water levels reduction is in the order of (0.43-1.02) m. As the floodpulse is essential for the ecological habitats, fisheries and livelihoods of the region, its reduction will pose major implications throughout the basin, from the Tonle Sap system to the Vietnamese Mekong Delta downstream.

## 1 Introduction

The Mekong River in Southeast Asia has attracted much attention as water infrastructure developments have accelerated in the past years (Best, 2019; Soukhaphon et al., 2021). Due to its transboundary nature, the cross-border hydrological impacts of anthropogenic alterations have become a contentious topic for states and agencies (Stone, 2010). Generally, developments can be separated into three geographical areas: 1. The Upper Mekong Basin (UMB) or the Lancang Basin in China, 2. The stretch from Chiang Saen in Thailand to Stung Treng in Cambodia and 3. The stretch from Stung Treng to the Vietnamese

Mekong Delta (VMD) that also includes the Cambodian floodplains and the Tonle Sap system.

In the UMB, the Lancang cascade consisting of 11 dams over an 800m drop was built by China beginning with Manwan Dam in 1992 (Hecht et al., 2019). These dams have raised concerns due to their ability to alter the hydrological regime downstream (Lu et al., 2014a). For example, the two largest, Xiaowan and Nuozhadu, completed in 2010 and 2014 respectively, has a total reservoir capacity of 38.3$km^3$, which is more than half of the total capacities of all the reservoirs in





the whole Mekong basin (MRFI, 2020). The cascade has been found to increase dry-season discharge and reduce wet-season discharge downstream (Li et al., 2017; Räsänen et al., 2012) to as far as Kratie (Räsänen et al., 2017).

Along the stretch from Chiang Saen to Stung Treng, a series of 11 hydropower dams are being planned by Laos and Cambodia (Grumbine and Xu, 2011). The first mainstream dam – the 1285MW Xayaburi – was operationalised recently in 2019. In addition, the area also encompasses smaller dams and irrigation infrastructures along its major tributaries such as
the Chi-Mun River and the 3S (Sekong, Sesan, Srepok) Rivers. At the Chi-Mun system, Cochrane et al. (2014) have pinpointed the Pak Mun Dam as a significant regulator of downstream Mekong flows. Similarly, the dams in the 3S basin have also been shown to alter hydrological conditions along the mainstream Mekong (Arias et al., 2014; Piman et al., 2013a).

Further downstream, the Cambodian floodplains and the Tonle Sap Lake system is home to a unique geographical
phenomenon. During the dry season, the Lake empties into the Mekong. However, during the wet season, large tracts of the floodplains are inundated, and flow is reversed from the Mekong to the Lake. This annual flood pattern is critical for both fisheries productivity (Halls and Hortle, 2021; Sabo et al., 2017; Ziv et al., 2012) and the agrarian communities that are reliant upon the annual floodwaters for replenishment of nutrients and water (Arias et al., 2012; Grundy-Warr and Lin, 2020).

Due to the significance of the annual floods, the ecological and hydrological services provided by the Cambodian floodplains is best understood as a consequence of the floodpulse (Junk et al., 1989). Elsewhere, hydrological alterations to the floodpulse have been quantified in the Amazon basin (Zulkafli et al., 2016) and Missouri basin (Bovee and Scott, 2002). Within other parts of the Mekong, the floodpulse has been investigated vis-à-vis its relationships to climate (Räsänen and Kummu, 2013; Västilä et al., 2010) or ecosystems (Arias et al., 2013; Kong et al., 2017; Ngor et al., 2018). These studies
indicate that quantification of the floodpulse can be useful in the understanding of the hydrology of a floodplain system.

In the Cambodian floodplains, water levels in the Tonle Sap Lake and the Mekong mainstream have a  close relationship (Guan and Zheng, 2021; Inomata and Fukami, 2008), meaning that any alterations to flows on the mainstream will affect the hydrology on the Lake. Studies on the Tonle Sap Lake has predicted dry-season water levels to increase and wet-season water levels to decrease (Arias et al., 2012; Kummu and Sarkkula, 2008). Indeed, remotely-sensed data has confirmed that
the surface area of the Lake has been on a declining trend since 2000 (Ji et al., 2018; Lin and Qi, 2017)

Overall, the Cambodian floodplains can protect neighbouring Phnom Penh and the VMD downstream from floods by storing large volumes of water during the wet season (Fujii et al., 2003; MRC et al., 2004). In addition, the effects of upstream water infrastructure development, such as the Lancang dams, on the VMD are also dampened by the buffering effect of the floodplains (Dang et al., 2005). However, despite its importance, the hydrological changes experienced by the Cambodian
floodplains over the past decades have been poorly understood.

Given such concerns, this study is structured with two aims. First, we aim to quantify the floodpulse along the Cambodian floodplain using sixty years of data from 1960-2019. Second, we aim to identify anthropogenic factors that have caused the alterations to the floodpulse. In so doing, we present the implications of current human activities on the Cambodian floodplains and the wider region.





## 2 Study Area

Beginning from Stung Treng, the lower reaches of the Mekong winds through a large floodplain that is seasonally inundated during the wet season (Figure 1). Connected to the floodplain is the Tonle Sap system that expands from an area of $2500\text{km}^2$ during the dry season to up to $15\,000\text{km}^2$ during the wet season (Ji et al., 2018). The whole floodplain is underlain by a mosaic of tropical ecosystems such as gallery forest, shrublands and aquatic herbaceous vegetation(Araki et al., 2007; Arias et al., 2013; Kummu and Sarkkula, 2008), and man-made landuse such as rice fields and canals (Mahood et al., 2020; Olson and Morton, 2018).



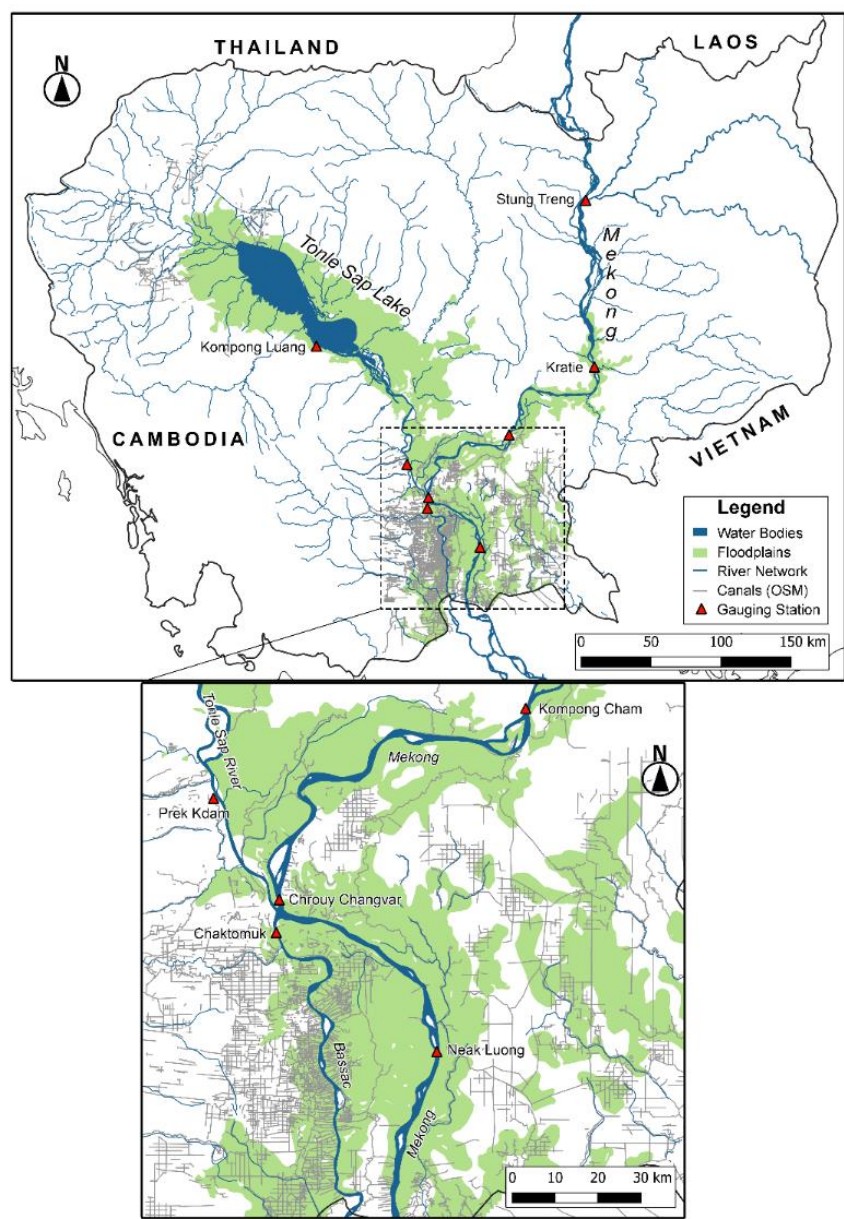

**Figure 1. Map of Cambodian floodplains. Key hydrological stations referenced in the paper are highlighted. Canals in grey are extracted from OpenStreetMap and copyrighted to OpenStreetMap contributors 2021. Distributed under the Open Data Commons Open Database License (ODbL) v1.0.".**


When water level exceeds 12m at Kompong Cham, extensive overbank flooding will occur on both banks of the Mekong (Inomata & Fukami, 2008; MRC et al., 2004). Concurrently, along the Tonle Sap River, water flows from the Mekong to the Tonle Sap Lake, bringing along its supply of sediments and nutrients (Campbell et al., 2009; Lu et al., 2014; Siev et al., 2018). In recent years, the Cambodian floodplains have been developed to tap into its potential for rice production (Erban





and Gorelick, 2016; Yu and Fan, 2011). Under the "Rice-White Gold" policy under the Cambodian government, the country fulfilled its objective to increase rice yield to 4 million tons in 2015 from just 20 thousand tons in 2009 (Royal Government of Cambodia, 2010).

The boom in rice production was possible with both the use of high-yielding rice varieties and an expansion of irrigation infrastructure that allowed multiple cropping (ADB, 2019). With investments from international donors such as the World
Bank and the Asian Development Bank (ADB) or countries such as Japan and China, Cambodia has both upgraded its ageing irrigation schemes and constructed new canals and reservoirs (Sithirith, 2017) (Table S1 in Supporting Information). However, the hydrological implications of these large-scale irrigation projects have not been accessed.

Another anthropogenic activity that has increased pace is sand-mining within the river channels (Kondolf et al., 2018; Schmitt et al., 2017). In Cambodia alone, it was estimated that at least 34.4 million cubic meters of sediments were mined
per year (Bravard et al., 2013). Of this amount, 18.1 million cubic meters was mined within the short stretch of Mekong from Kompong Cham to the Vietnamese border (Bravard et al., 2013). As the large volumes of sediment are being removed from the river bed, sediment contribution from the Upper Mekong is unable to replenish these losses (Hackney et al., 2020). The problem is so severe in the neighbouring VMD that sand-mining has resulted in up to 1.3m decrease in water levels, exacerbating the impacts of sea-level rise or saltwater intrusion (Brunier et al., 2014; Vu et al., 2018). Thus, if sand-mining
in Cambodia has also altered the channel morphology of the Mekong, then we will expect a resultant change to the channel hydrology.

## 3 Materials and Methods

### 3.1 Data

Water level and discharge data are obtained from the Mekong River Commission (MRC, 2021). On the Mekong mainstream,
we get data from Stung Treng, Kratie, Kompong Cham, Chrouy Changvar and Neak Luong. We also obtain data from Prek Kdam (Tonle Sap River), Chaktomuk (Bassac River) and Kompong Luang (within Tonle Sap Lake) These stations are selected due to their good coverage of the Cambodian floodplains and documentation of their historical records from 1960 onwards (Table 1).

Gaps in discharge data at Neak Luong from 2012-2019 and Chaktomuk from 2012-2019 are calibrated from its
corresponding water levels values using these rating curves derived from MRC et al. (2004):

$$\text{Neak Luong (NL): } Q_{NL} = (12.718H_{NL} + 62.250)^2(H_{CC} - H_{NL})^{0.2}$$

$$\text{Chaktomuk (CK): } Q_{CK} = (13.943H_{CK} + 19.992)^{1.8}$$

where Q is discharge and H is water levels. $H_{CC}$ refers to water level at Chrouy Changvar. These rating curves were based on 70 measurements taken from July 2002 to October 2003 (MRC et al., 2004).

In addition, we identify an unreported change in rating curve at Stung Treng that depressed water discharge readings post-2005 (Lu and Chua, 2021). These discharge values are calibrated with the following relation obtained from the 2000-2004 rating curve:



Stung Treng (ST): $Q_{ST} = 207.549 H_{ST}^2 + 2598.316 H_{ST} - 4854.477$ (R$^2$>0.999)

**Table 1. Summary of obtained data records from the various stations**

| | | Obtained Data | | Calibrated Data | Treatment |
|---|---|---|---|---|---|
| | | Water Level | Discharge | | |
| Mekong mainstream | Stung Treng | 1960-2019 | 1960-2019 | 2005-2019 | Calibrated to 2000-2004 rating curve because new rating curve depressed discharge values from 2005 onwards. |
| | Kratie | 1960-2019 | | | |
| | Kompong Cham | 1960-2019 | | | |
| | Chrouy Changvar | 2012-2019 | | | |
| | Neak Luong | 1960-2019 | 1960-2011 | 2012-2019 | Used the rating curve from MRC et al. (2004) to estimate discharge for missing years |
| Tonle Sap River | Prek Kdam | 1960-2019 | 1962-1972 1995-2018 | | |
| Bassac River | Chaktomuk | 1960-2019 | 1960-2011 | 2012-2019 | Used the rating curve from MRC et al. (2004) to estimate discharge for missing years |
| Tonle Sap Lake | Kompong Luang | 1996-2019 | | | |


To compare the changes in present and past hydrological conditions of the Cambodian floodplains, data are divided into three timeframes: 1960-1991, 1992-2009 and 2010-2019. These three timeframes represent distinct phases of water infrastructure development. 1960-1991 constitutes the pre-dam era and can be treated as the historical baseline before any major water infrastructure development. Following the construction of the first dam on the Mekong mainstream – Manwan

Dam, the growth era from 1992-2009 has seen extensive hydropower development both on the Mekong mainstream and its tributaries upstream in China, Thailand, and Laos. For example, during this period, China constructed the Manwan, Dachaoshan and Jinghong dams in the Upper Mekong Basin, with a total storage capacity no less than 2.95km$^3$ (MRFI, 2020). After 2010, the pace of dam construction increased with the operationalisation of mega-dams in China such as Xiaowan and Nuozhadu dams with a combined storage capacity of up to 38.3km$^3$, thereby marking the start of the mega-dam

era (MRFI, 2020). Additionally, the Cambodian government announced the "Rice-White Gold" policy paper in 2010, sparking a burst of intensive irrigation projects in the Cambodian floodplain (ADB, 2019a; Royal Government of Cambodia, 2010).

### 3.2 Methodology

#### 3.2.1 Floodpulse

The various parameters to characterise floodpulse as seen in Figure 2a are determined from the equations below:

$MIN_{annual}$/m = lowest water level of a year





$MAX_{annual}$/m = highest water level of a year

$AMPLITUDE$/m = $MAX_{annual}$ - $MIN_{annual}$

$THRESHOLD\ (FT)$/m = 50$^{th}$ percentile of all water levels in study period

$START\ DATE$ = Date when water level > $FT$ AND remain more than $FT$ for next 10 days

$END\ DATE$ = Date when water level < $FT$ AND remain less than $FT$ for next 10 days

$DURATION$/days = Days between $START\ DATE$ and $END\ DATE$

$PEAK\ DATE$ = Date of $MAX_{annual}$

$$RISE\ RATE/\text{mday}^{-1} = \frac{MAX_{annual} - FT}{Days\ from\ START\ DATE\ to\ PEAK\ DATE}$$

$$FALL\ RATE/\text{mday}^{-1} = \frac{MAX_{annual} - FT}{Days\ from\ PEAK\ DATE\ to\ END\ DATE}$$





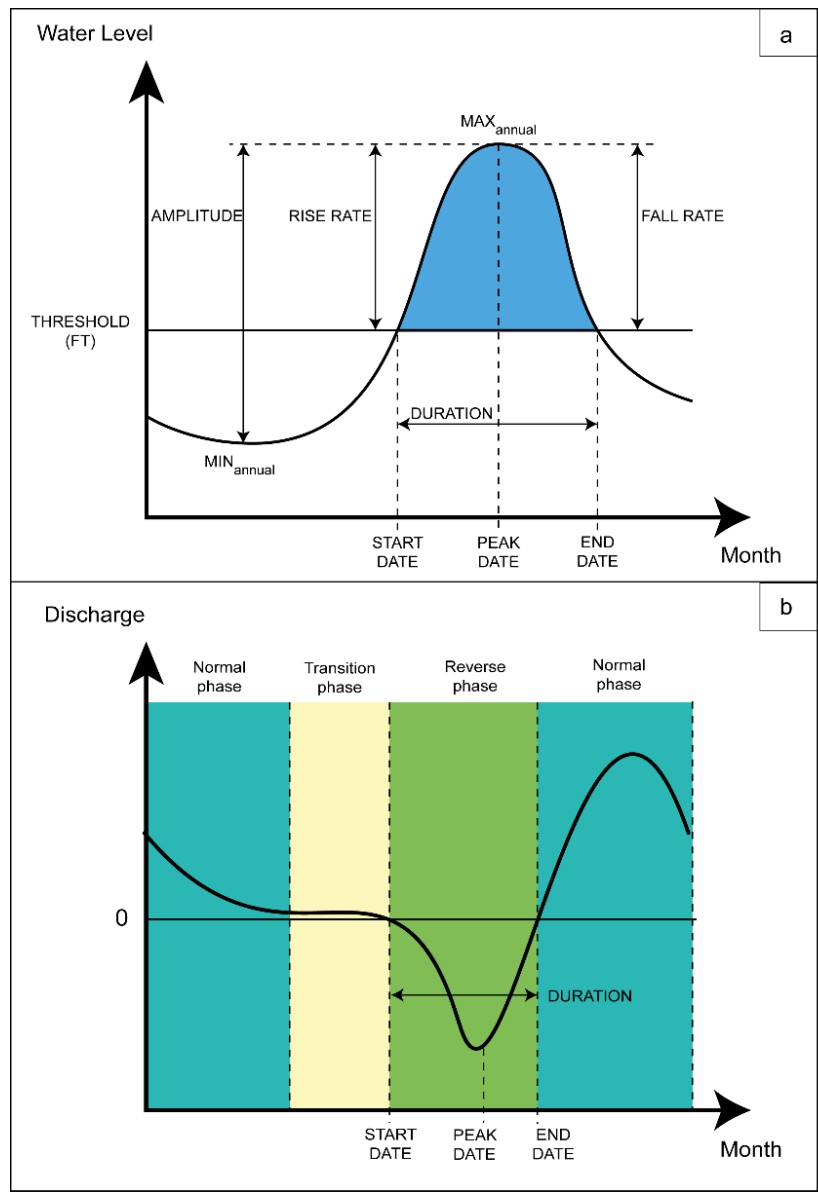

**Figure 2a (top). Schematic of floodpulse variables in a typical annual water level plot in the Cambodian floodplains. Figure 2b (bottom). Schematic of the various phases at Prek Kdam observed through a typical annual discharge plot. Negative discharge values indicate reverse flow from the Mekong to the Tonle Sap Lake.**

Comparisons of the means of numerical variables such as *AMPLITUDE*, *RISE RATE* and *FALL RATE* are validated with Welch's t-test. For comparison of dates, the date is first converted to its Julian date. Thereafter, the median date of each study period is compared and validated with a Mann-Whitney test.





### 3.2.2 Reverse Flow at Prek Kdam

The parameters to characterise reverse flow (RF) where water flow from the Mekong to Tonle Sap Lake is stated below:

$$\text{ANNUAL RF/km}^3 = \sum RF \text{ in a year}$$

$$\text{MAX}_{RF} \text{ /cms} = \text{highest RF of a year}$$

Three phases of the flow pattern are observed: Normal phase, transition phase and reverse phase (Figure 2b). The normal phase is when water flow from the Tonle Sap towards the Mekong. The transition phase begins on the date when the river record water flowing in the reverse direction for the first time in the year. The reversal phase starts when water continues to
flow towards the lake and does not switch direction until the water abruptly changes back to the normal phase flow. Note: not all years have a transition phase.

### 3.2.3 Water loss/gain in the Cambodian Floodplains

Difference in discharge ($Q_{diff}$) between the downstream stations of Chaktomuk (CK) and Neak Luong (NL), and the upper station of Stung Treng (ST) is calculated to estimate the amount of water loss/gain in the Cambodian floodplains.

$$Q_{diff} = (Q_{CK} + Q_{NL}) - Q_{ST}$$

If $Q_{diff} = 0$, then it means that the amount of water entering the floodplain system is roughly equivalent to the amount exiting. If $Q_{diff} > 0$, then it means that there is additional water to the Cambodian floodplains from outside the Mekong mainstream. This addition can come from precipitation or discharge from tributaries. Conversely, if $Q_{diff} < 0$, then it means that water is lost to outside the Cambodian floodplains through diversion or evapotranspiration.

## 4 Results

### 4.1 Mekong River mainstream

#### 4.1.1 Changes to annual flood extent

The annual flood extent is given by the yearly maximum and minimum water levels of the Mekong. Compared to the pre-dam era from 1962-1991, minimum water level was higher during the growth era from 1992-2009 (Figure 3). The trend
continued into the mega-dam era (2010-2019) with significant increases recorded at all stations except Chaktomuk (Table S2 in supporting information). For reference, the minimum water levels were higher by 0.60m at Stung Treng; 1.55m at Kratie; 0.60m at Kompong Cham and 0.10m at Neak Luong.

Furthermore, maximum water levels have decreased at all stations downstream of Kratie during the mega-dam era as compared to the pre-dam era. For example, Neak Luong and Chaktomuk experienced 0.55m and 0.76m drops respectively.
Correspondingly, amplitude of the floodpulse decreased. Comparing records from the mega-dam era to the pre-dam era, amplitude dropped by 7.9% at Kratie; 5.6% at Kompong Cham; 10.6% at Neak Luong; and 8.9% at Chaktomuk. This observed increase in dry-season minima and decrease in wet-season maxima is consistent with studies in other parts of the





Mekong Basin (Binh et al., 2020b; Li et al., 2017; Räsänen et al., 2017), demonstrating that the impacts of water infrastructure development are evident within the Cambodian floodplains.

In practical terms, more areas of the Cambodian floodplains are now permanently inundated during the dry season. As the flood amplitude decreases, the annual flood extent is reduced, meaning that some parts of the floodplains are no longer being flooded during the wet season.

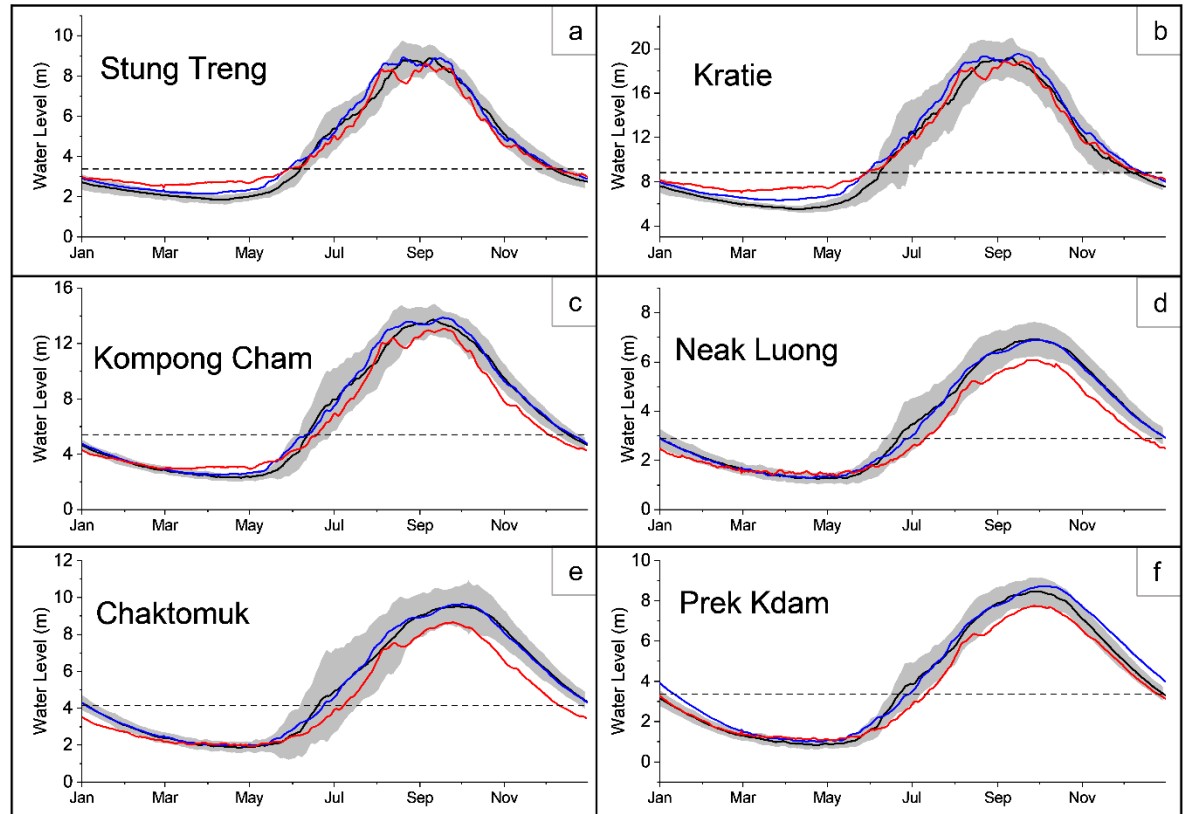

**Figure 3. Variation of mean water levels at various stations within the Cambodian floodplains over time. (Dotted horizontal line:**
**flood threshold; black curve: water levels from 1960-1991; blue curve: water levels from 1992-2009; red curve: water levels from 2010-2019. Shaded region represents 1 standard error of mean water levels from 1960-1991)**

**4.1.2 Changes to annual flood duration**

When compared against 1960-1991 records, the flood seasons in 1992-2009 did not show many significant changes. However, the mega-dam era showed greater alterations in flood timing. Downstream from Kratie, flood duration decreased
significantly by 26 days at Kompong Cham; 36 days at Neak Luong; and 40 days at Chaktomuk. The drastic shortening of the flood season by up to more than a month was caused by both a delay in floodpulse and an early end to the flooding. The delay in the start of the flooding season was observed to increase with further distance downstream. For instance, while the start date was only later by 9 days at Kompong Cham, it was later by 11 days at Chaktomuk and 15 days at Neak Luong. For





the end dates of the flood season, it was earlier by 18 days at Kompong Cham; 25 days at Chaktomuk; and 18 days at Neak Luong. Thus, for areas downstream of Kratie, the floodwaters have indeed arrived later and receded earlier, resulting in a shorter wet season and a longer dry season.

### 4.1.3 Changes to rise and fall rates

At Kompong Cham, Chaktomuk and Neak Luong, both rise rates and fall rates were observed to increase during the mega-dam era, as compared to the pre-dam era. The largest change in rise rate was observed at Chaktomuk with a 53.8% increase
from 0.062m/day to 0.096m/day. In terms of fall rates, the largest percentage change was observed at Kompong Cham: a 23.8% increase from 0.089m/day to 0.111m/day. As rise and fall rates are indicators of upstream human modification (Cochrane et al., 2014; Richter et al., 1997), the alterations hint at anthropogenic hydrological regulation in the region.

### 4.2 Tonle Sap System

### 4.2.1 Changes to water exchange at Tonle Sap River

At Prek Kdam, the hydrological changes mirrored that in the Mekong mainstream. Compared against pre-dam records, the mega-dam era saw minimum water levels increased significantly by 0.25m and maximum water level decreased by 0.79m. This resulted in a significant 12.9% reduction in amplitude – from 8.27m to 7.20m. Furthermore, the flood duration was shorter by around 20 days. The average start date in the mega-dam era was 10th July, a significant 15-day delay from 25th June previously during the pre-dam era.

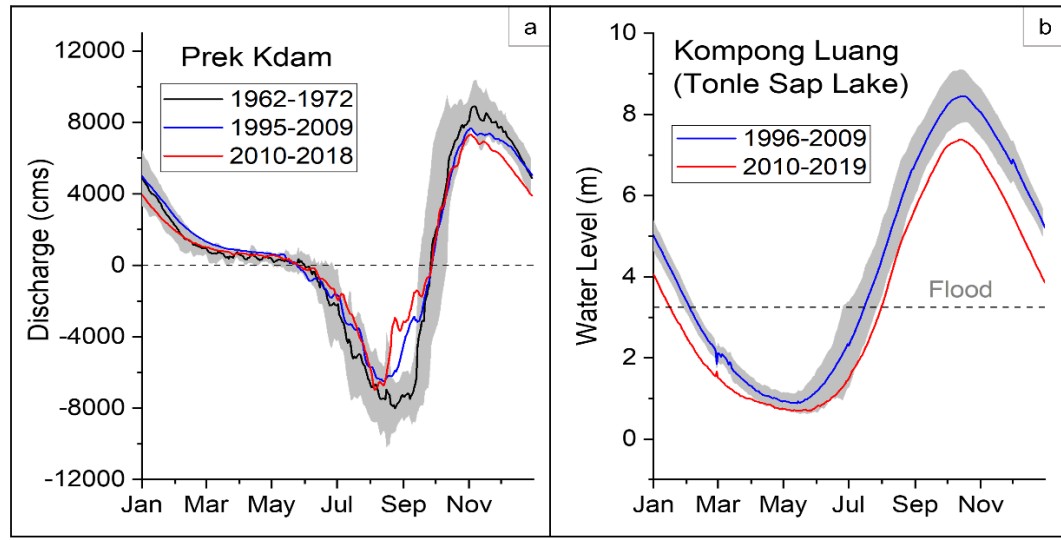


**Figure 4a (left). Change in discharge at Prek Kdam over time. Shaded region indicates standard error of mean discharge during 1962-1972. Figure 4b (right). Change in water level at Kompong Luang over time. Shaded region indicates standard error of mean water level during 1996-2009.**





### 4.2.1 Changes to Tonle Sap Lake flooding pattern

At the Kompong Luang station in the Tonle Sap Lake, annual minimum water levels decreased from 0.70m in 1996-2009 to 0.60m in 2010-2019 (Table S4 in Supporting Information). Similarly, yearly maximum water level dropped significantly from 8.58m (1996-2009) to 7.52m (2010-2019). Using the water level (H) to volume (V) and area (A) relation as derived by Kummu et al. (2014),

$$A = -5.5701H^3 + 137.40H^2 + 470.29H + 1680.2$$
$$V = 0.7307H^2 - 0.3554H + 0.9127$$


the 1.05m drop in maximum water level meant that 3990 km$^2$ of previously seasonally inundated land is now permanently dry. Correspondingly, the reduction in maximum water volume is a drastic 12.1 km$^3$. This reduction translates to a decrease in 20.6% of maximum area and 23.4% of maximum water volume. During the dry season, the minimum area is now 3.1% smaller (2080 km$^2$ to 2010 km$^2$) and contain 5.8% less water (1.02 km$^3$ to 0.96 km$^3$).

This observed decrease in both minimum and maximum water levels is consistent with recent scholars who monitored the change in Tonle Sap Lake area using remote sensing methods (Ji et al., 2018; Lin and Qi, 2017; Wang et al., 2020). However, the observed decrease in dry season flow does not follow hydro-modelling results that suggest that water level should be higher during the dry season at the Tonle Sap Lake (Arias et al., 2012, 2014; Kummu and Sarkkula, 2008; Piman et al., 2013b). We postulate that the models adopted might not have considered the development of water infrastructures at 235 the Tonle Sap Lake tributaries. For instance, reservoirs are being developed at Stung Sreng and Stung Pursat —two key tributaries of the Lake (ADB, 2019a). These irrigation projects could have decreased dry season flow to the lake, resulting in the decrease of dry season water level in Tonle Sap Lake.

Additionally, there was a significant drop in flood duration, from 198 days to 163 days. This reduction was caused by an early end to the flood season. During 1996-2009, the average end date of the flood season was 31$^{st}$ January but during 2010-240 2019 the season ended roughly on 18$^{th}$ January, earlier by 13 days. The shortened flooding season follows the same trend as in the Mekong mainstream.

## 5 Discussion

### 5.1 Reduction of the floodpulse

Figure 5 shows a schematic of the annual floodpulse on the Cambodian floodplains. During the pre-dam era of 1960-1991, 245 discharge at Stung Treng during the wet season was 25 500cms. Downstream at Kratie, discharge reduced slightly to 24 000cms which could be caused by overland flooding between Stung Treng and Kratie. From Kratie to Kompong Cham, as additional water arrived from the surrounding watersheds, discharge increased to 27 100cms. Thereafter, extensive overland flooding was experienced at the Chaktomuk confluence of Mekong, Bassac and Tonle Sap Rivers. At Prek Kdam, 4270cms of water flowed to the Tonle Sap Lake. At Chaktomuk and Neak Luong, 2800cms and 16 400cms of water continued to flow 250 towards the Vietnamese Delta via the Bassac and Mekong Rivers respectively.





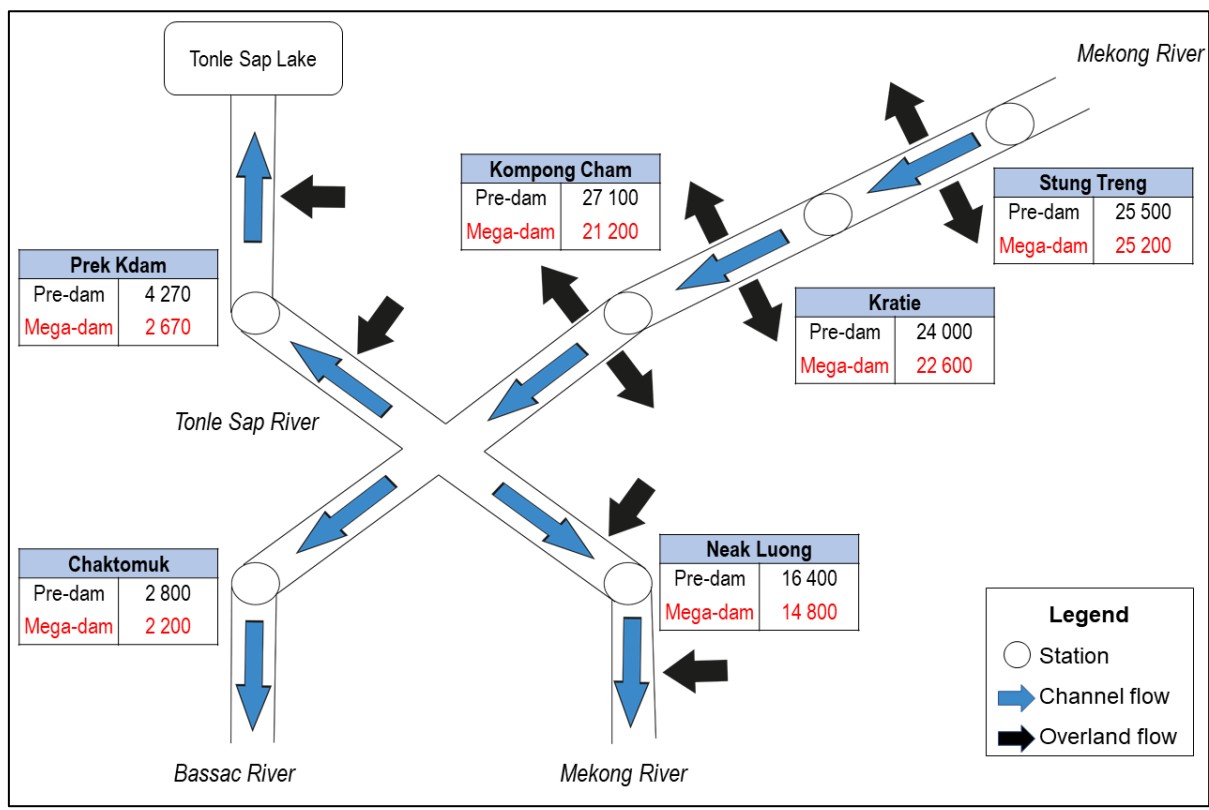

**Figure 5. Schematic of wet season discharge on the Cambodian floodplains during the pre-dam and mega-dam era. Across all stations, there is a reduction of discharge during the mega-dam era of 2010-2019.**

During the mega-dam era of 2010-2019, there was a reduction in wet-season discharge across all stations. Since Stung
Treng, the uppermost station of the Cambodian floodplains, only registered a reduction of 300cms, then the cumulative contribution of infrastructure and precipitation in the upper reaches of the Mekong only accounted for 300cms of discharge. As seen in Table 2, discharge at Kratie and Kompong Cham reduced to 22 600cms and 21 200cms respectively, of which only 21% (Kratie) and 5% (Kompong Cham) could be attributed to the upper reaches. Similarly, Prek Kdam experienced a decrease in discharge of -1 600cms; Neak Luong: -1 600cms and Chaktomuk: -600cms. Together, only about 8% of the flow
reduction at the three lower stations could be attributed to developments upstream of Stung Treng.






**Table 2. Percentage contribution of influences upstream of Stung Treng to the reduction of discharge at the various stations in the Cambodian floodplains.**

|  | Wet-season discharge/cms | | From upper reaches/cms | % contribution | From Cambodian floodplains/cms | % contribution |
|  | Pre-dam | Mega-dam |  |  |  |  |
| --- | --- | --- | --- | --- | --- | --- |
| Stung Treng | 25 500 | 25 200 | -300 | 100% |  |  |
| Kratie | 24 000 | 22 600 | -300 | 21% | -1 100 | 79% |
| Kompong Cham | 27 100 | 21 200 | -300 | 5% | -5 600 | 95% |
| Downstream (Prek Kdam + Neak Luong + Chaktomuk) | 23 470 | 19 670 | -300 | 8% | -3 500 | 92% |

$Q_{diff}$ provides a reference to the amount of water entering or leaving the Cambodian floodplains system. As seen in Figure 6, whether $Q_{diff}$ is positive or negative depends on the time of the year. In the dry months from October to April, $Q_{diff}$ is generally positive, implying that there is a net contribution of water from the tributaries. Alternatively, in the wet months from May to September, $Q_{diff}$ is generally negative as water is lost through the increased evapotranspiration from the flooding. This alternating gain-loss pattern is elucidated in Figure 7.





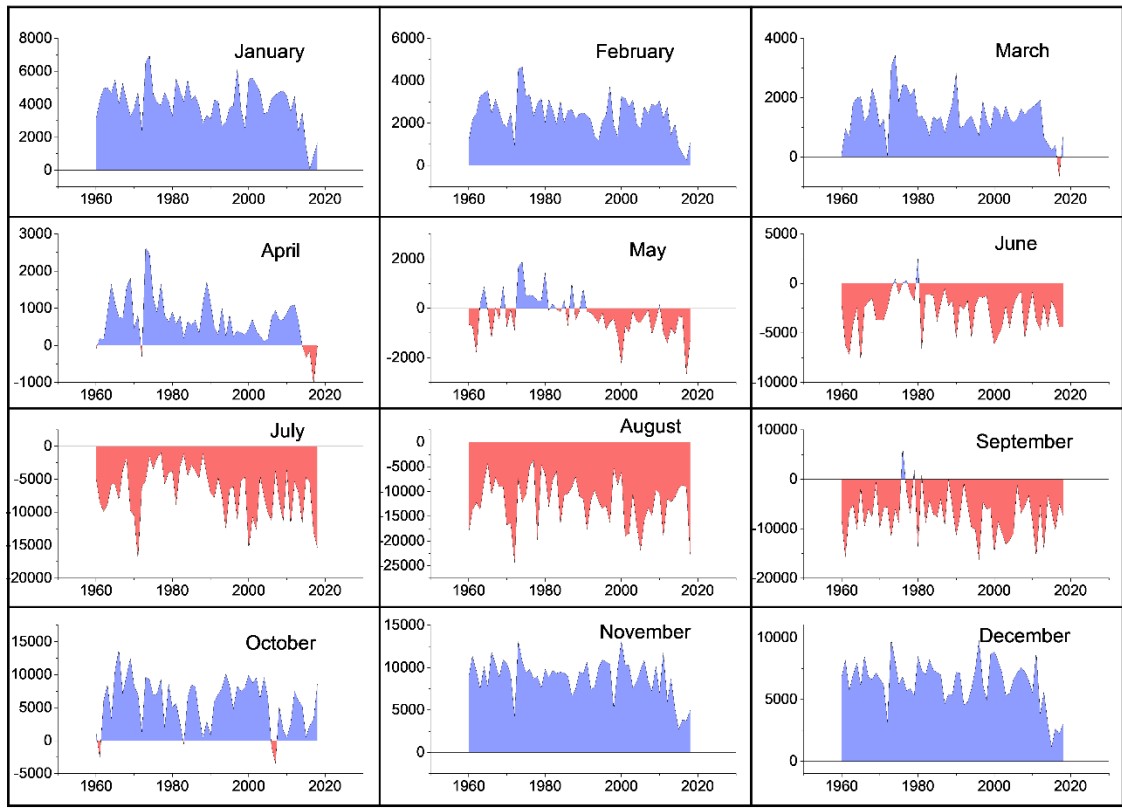

**Figure 6. Monthly plots of $Q_{diff}$/cms from 1960-2019. Blue zones indicate $Q_{diff}$>0, meaning an addition of water from outside the Cambodian floodplains. Red zones indicate $Q_{diff}$<0, meaning a loss of water to outside the Cambodian floodplains. Generally, during the wet-season from June to September, $Q_{diff}$<0.**



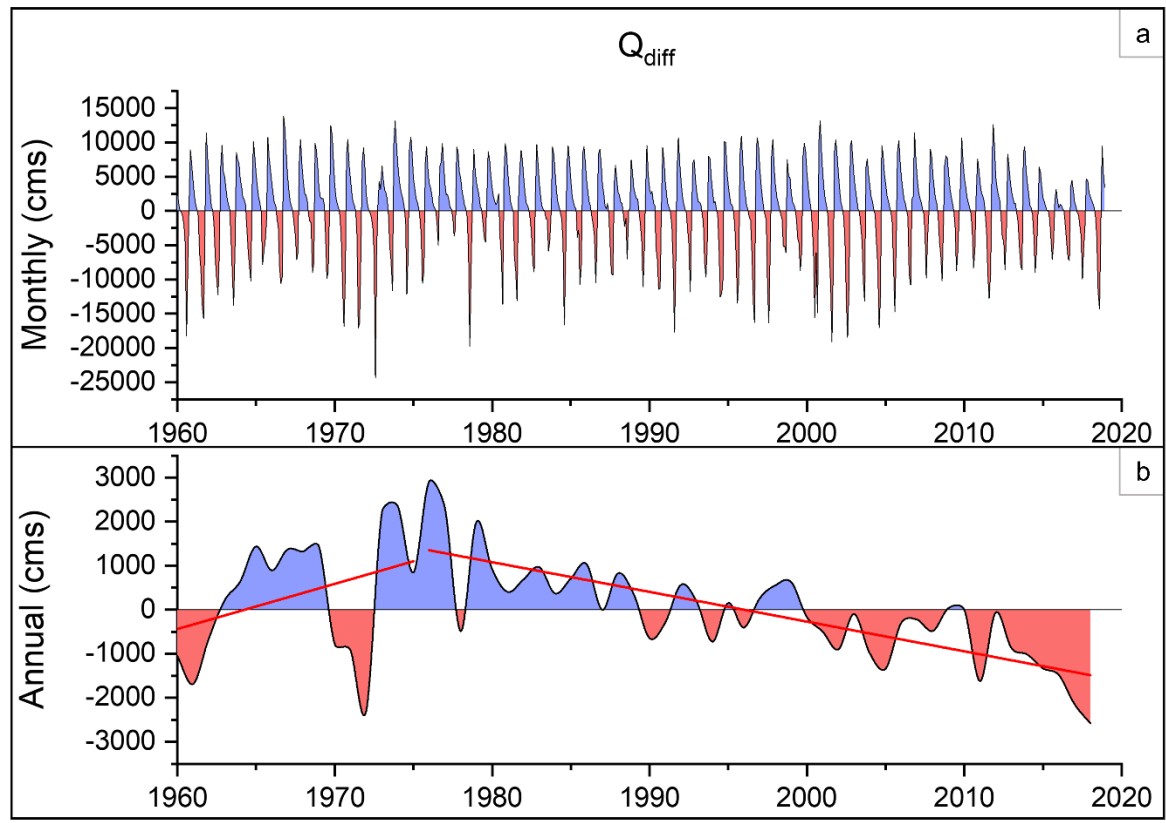

**Figure 7a (top). Time-series plot of $Q_{diff}$ from 1960 to 2019 with monthly steps. Figure 7b (bottom). Time-series plot of $Q_{diff}$ from 1960 to 2019 with yearly steps. From 1976-2019, $Q_{diff}$ decreased at a slope of -(68 ± 8) cms/yr (p<0.01).**

Figure 7b also showed two general trends in $Q_{diff}$ across time. During 1960-1975, $Q_{diff}$ was on an increasing trend. However, from 1976-2019, $Q_{diff}$ decreased significantly at a slope of -(68 ± 8) cms/yr. This implied that the Cambodian floodplains have been losing water at a rate of (2.1 ± 0.3) km$^3$/yr. The only natural mechanism able to explain the reduction in $Q_{diff}$ is that overbank flooding has become more frequent, leading to increased loss of water through evapotranspiration. However, from the analysis of floodpulse in Section 4.1, we have identified that both flooding extent and duration is on a decreasing trend. Therefore, the only plausible mechanism for the drop in $Q_{diff}$ is man-made diversion of Mekong flows.

While much attention has been given to the role of dams in the upper reaches of the Lower Mekong Basin (Lu and Siew, 2006; Räsänen et al., 2012) or tributary basins such as the 3S rivers (Arias et al., 2014), they are unlikely to be the main factors influencing the floodpulse in the Cambodian floodplains. At Stung Treng – the start of the Cambodian floodplains – there were no significant changes to its maximum flood levels or its flood duration. Also, as observed in Figure 8, measured rainfall in the Cambodian floodplains has remained roughly constant from 1960-2019, in line with observations via other sensing methods (Raghavan et al., 2018; Singh and Qin, 2020; Thoeun, 2015). Thus, the observed reduction of flood discharge in the Cambodian floodplains cannot be attributed solely to either upstream developments or natural climatic variability – local anthropogenic factors are likely the main reason.





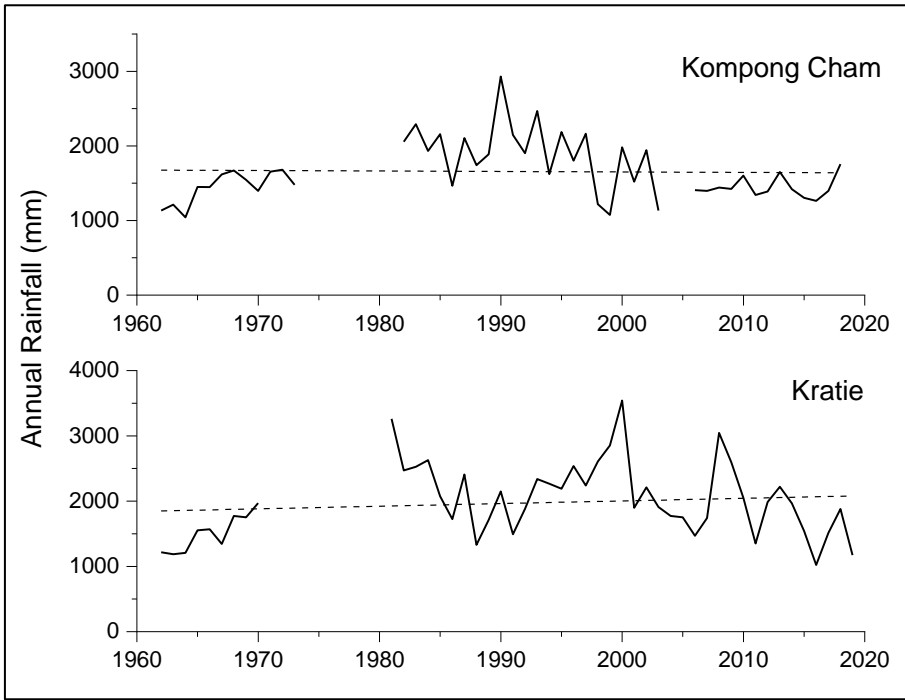


**Figure 8. Measured precipitation data at Kompong Cham and Kratie from 1960-2019. At both stations, there are no statistically significant trends in rainfall within the data period.**

## 5.2 Impacts of water regulation

The scale of irrigation and construction of irrigation reservoirs has been increasing since the Cambodian government
announced its "Rice-White Gold" policy with numerous large projects. For instance, reservoirs have been constructed along the tributaries feeding to Tonle Sap Lake. Further downstream, projects such as the Vaico irrigation project has seen canals constructed across the floodplains east of the Mekong. These infrastructures draw water from the Mekong, resulting in the observed decrease in $Q_{diff}$.

As Figure 6 shows, there has been a sharp decline in $Q_{diff}$ from November to April from about 2010 onwards, coinciding
with the start of the Cambodian "Rice-White Gold" policy. Indeed, the dry-season reduction is consistent with irrigation operations as water is diverted to the dry fields. With the additional water, farmers could increase their dry-season yield that had remained low due to the lack of water (Kea et al., 2016).

While irrigation coverage has increased to approximately 22.6% of paddy land by 2015, the percentage is still much lower than neighbouring Thailand and Vietnam (Kea et al., 2016), meaning that Cambodia can still expand its rice production. As
drought occurrences are likely to increase in the future (Oeurng et al., 2019), water availability will be the major challenge for the continued expansion of the Cambodian agriculture sector (Bresney et al., 2020; Sithirith, 2021).

However, the large irrigation schemes might not be effective as they often require high maintenance (IWMI-ACIAR, 2013). For instance, some reservoirs are located at lower elevations than the fields, thereby limiting the usability of the canals





(Sareth et al., 2020). Also, these large-scale projects are limited in reaching smaller-scale holdings, where smaller-scale
canals will be more efficient (Sithirith, 2017). For example, the Chinese-funded Vaico Project consisting of 73km of canals
has been accused of running dry before reaching the fields (Blake, 2018). Given such problems, it appears that there is much
wastage of water from the river to the fields.

**5.3 Impacts of channel incision**

Due to hydropower developments in the Upper Mekong Basin, sediment flux has declined in the past decades (Kondolf et
al., 2014; Kummu et al., 2010; Lu et al., 2014; Wang et al., 2011). Concurrently, sand-mining activities have further
removed sediments from the riverbed (Hackney et al., 2020). Together, the reduction in sediments has resulted in greater
erosion of the channels, leading to channel deepening (Kondolf et al., 2018). To quantify the effects of channel incision on
the flood pulse, we analysed water levels at downstream Chaktomuk and Neak Luong where sand-mining has been the most
active (Bravard et al., 2013).

We estimate the contribution of sediment decline and irrigation to the reduction of water levels by comparing water level
changes at the start and end of the flood season. At the start of the flood season in July, minimal water will be diverted for
irrigation as the fields will be flooded naturally. Thus, changes in water levels in July will be caused predominantly by
incision. Conversely, at the end of the flood season in December, water from the receding floodwaters will be diverted for
irrigation in anticipation of the dry season. Therefore, the changes in water levels in December will reflect influences from
both irrigation and incision.

In the absence of man-made hydrological alterations, water levels at upstream Kratie will be tightly coupled with those at
downstream stations throughout the year. However, as seen in Figure 9, water levels at Neak Luong and Chaktomuk have
been decreasing with respect to water levels at Kratie. In other words, for the same water level at Kratie, the water level at
Neak Luong/Chaktomuk has decreased from the 1960s to 2010s. Table 3 indicates how much the water level has dropped in
the growth period and mega-dam eras.

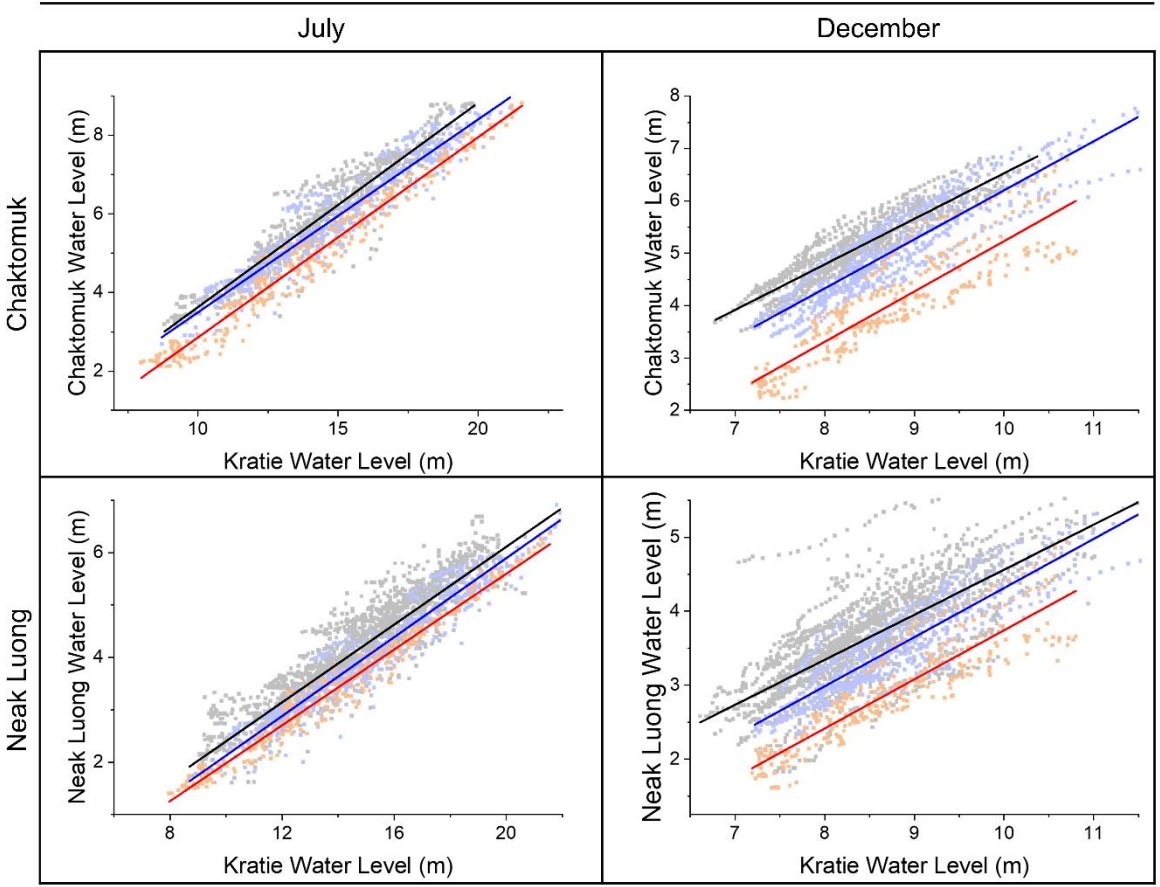

**Figure 9. Comparison of water levels at Neak Luong/Chaktomuk and water levels at Kratie. (Black: 1960-1991; blue: 1992-2009; red: 2010-2019) July and December represent the start and end of the flooding season respectively. To calculate how much water level has dropped at Neak Luong/ Chaktomuk over time, we calculate the mean vertical distance between the best-fit lines at 1960-1991, and 1992-2009/2010-2019.**





**Table 3. Water level (m) reduction at Chaktomuk and Neak Luong.**

| | Total water level (m) reduction | | | | Water level (m) reduction due to incision | | Water level (m) reduction due to water withdrawal (irrigation) | |
| --- | --- | --- | --- | --- | --- | --- | --- | --- |
| | July | | December | | | | | |
| | Growth | Mega-dam | Growth | Mega-dam | Growth | Mega-dam | Growth | Mega-dam |
| Neak Luong | 0.25 ± 0.02 | 0.46 ± 0.03 | 0.33 ± 0.05 | 0.90 ± 0.05 | 0.25 ± 0.02 | 0.46 ± 0.03 | 0.08 ± 0.07 | 0.44 ± 0.08 |
| Chaktomuk | 0.4 ± 0.1 | 0.96 ± 0.06 | 0.42 ± 0.06 | 1.42 ± 0.08 | 0.4 ± 0.1 | 0.96 ± 0.06 | 0.0 ± 0.2 | 0.5 ± 0.1 |

Note: Reduction during the growth era (1992-2009) or mega-dam era (2010-2019) are calculated with reference to the pre-dam era (1960-1991). Error margin represent 1 standard error from mean values.

At both Neak Luong and Chaktomuk, water level reduction was greater during the mega-dam era. Furthermore, both incision and irrigation have picked up pace during that period. For reference, while incision only caused $(0.25 \pm 0.02)$m and $(0.4 \pm 0.1)$m reduction in water levels at Neak Luong and Chaktomuk respectively during the growth era, it caused a larger $(0.46 \pm 0.03)$m (Neak Luong) and $(0.96 \pm 0.06)$m (Chaktomuk) drop during the mega-dam era. Similarly, reduction in water levels caused by water withdrawal during the growth era was $(0.08 \pm 0.07)$m at Neak Luong and $(0.0 \pm 0.2)$m at Chaktomuk. During the mega-dam era, water level dropped by $(0.44 \pm 0.08)$m at Neak Luong and $(0.5 \pm 0.1)$m at Chaktomuk. The results confirm the hypothesis that irrigation has played a bigger role in the alteration of the Cambodian floodplain floodpulse post-2010. Also, it appears that the problem of channel incision has gotten more severe.

**5.4 Wider environmental implications**

The Tonle Sap Lake has been decreasing in size throughout the years as shown in Section 4.2.2. A parallel can be drawn with Poyang Lake in Yangtze River. There, the Three Gorges Dam has reduced water level downstream. As water level was reduced, the hydraulic gradient from Poyang Lake to the Yangtze increased, resulting in a reduction of lake volume there (Zhang et al., 2014, 2015).

In the case of the Tonle Sap Lake, there is a reduction of water levels in the Mekong mainstream during the wet season. As expected from hydrological models (Inomata & Fukami, 2008; MRC et al., 2004), the reduced hydraulic gradient from the water levels between the Mekong and the Lake will lead to lesser water entering the Lake from the Mekong. During the pre-dam era, 74.54km$^3$ of water emptied into the Mekong during the normal flow phase while 49.67km$^3$ of water entered the Lake during the reverse flow phase annually, meaning that there was a net outflow of 24.87km$^3$. In the mega-dam era, only 68.81km$^3$ of water flowed into the Mekong during the normal phase and reverse flow decreased to 31.74km$^3$; net outflow from the Lake to the Mekong has increased to 31.07km$^3$. Summing up, this means that the Tonle Sap Lake has released





about 6.20km³ more water annually to the Mekong during 2010-2019 as compared to 1962-1972. Without a corresponding increase in inflows from either precipitation or the Lake tributaries, the Lake will decrease in volume over time.

Downstream of the Cambodian floodplains, the VMD will also be affected by the decreased floodpulse. While hydropower dam operations can increase dry season water levels (Dang et al., 2016), the combined effects of local channel incision (Binh et al., 2020b) and irrigation operations will reduce dry season water. Since irrigation infrastructure in the VMD is even more
developed than those in Cambodia (Tran and Weger, 2018), the impacts of dry-season extraction is likely to be even greater in the VMD. Therefore, the decreased floodpulse at Chaktomuk and Neak Luong is likely to propagate further downstream to the VMD. Also, a smaller floodpulse will bring fewer sediments to the VMD (Binh et al., 2021). As a result, current problems of land subsidence and seawater intrusion there may become more severe in the future (Binh et al., 2020a; Kantoush et al., 2017; Zoccarato et al., 2018).

The continued expansion of irrigation and sand-mining operations in the Cambodian floodplains may be unsustainable in the long run. The annual floodpulse is key to the regulation of the health of the floodplains, from fisheries to sediment replenishment. For instance, the annual floods increase soil health through buffering acidity and increasing its nutrients content (Dang et al., 2016; Sakamoto et al., 2007). A reduction of the floodpulse amplitude and duration will reduce the annual flood extent, thereby reducing the soil productivity. Concurrently, as the migration cycles of the fishes are intimately
tied to the flooding extent, the shift in floodpulse timing will affect the catch rates (Baran and Myschowoda, 2009; Enomoto et al., 2011).

## 6 Conclusion

This study demonstrates that the floodpulse at the Cambodian floodplains has indeed changed drastically in the past decade. Compared to 1962-1991 levels, minimum water levels in 2010-2019 has increased by 0.10m (Neak Luong) – 1.55m(Kratie).
As a result, flood amplitude has decreased by 5.6% (Kompong Cham) to 12.9% (Prek Kdam), meaning that the annual flood extent has reduced. Furthermore, the flood season has decreased by 26 days (Kompong Cham) – 40 days (Chaktomuk), with the flood season starting later and ending much earlier.

Correspondingly, the altered floodpulse along the Mekong mainstream affected the annual reverse flows along the Tonle Sap River. At Prek Kdam, total annual reverse flow dropped from 48.67 km³ in 1962-1972 to 31.74 km³ in 2010-2018,
representing a dramatic drop of 56.5%. Correspondingly, the Tonle Sap Lake has also been altered by this huge change. There, minimum and maximum water levels have dropped by 0.10m and 1.06m respectively. These reductions correspond to a decrease in Lake area of 3.1% in the dry season and 20.6% during the wet season.

We also identify two main reasons for these hydrological alterations – irrigation and sediment decline. The boom in irrigation infrastructure in the last decade has resulted in more water being diverted away from the Mekong and Tonle Sap
Lake to the fields. We estimate that an average of $(2.1 \pm 0.3)$ km³ of water is lost per year from the Cambodian floodplains from 1976-2019. In addition, declining sediments has caused incision to reduce water levels by $(0.46 \pm 0.03)$m at Neak Luong and $(0.96 \pm 0.06)$m at Chaktomuk in 2010-2019 when compared against 1960-1991 values. Together, irrigation and sediment decline has caused the observed reduction in floodpulse at the Cambodian floodplains.

As the hydraulic gradient governing the reverse flow to the Tonle Sap Lake decreases, the Lake may suffer a permanent
reduction in water volume. Furthermore, the impacts of the runoff reduction will be felt further downstream in the VMD.



This identified shift in floodpulse is non-trivial, with far-reaching ecological and environmental impacts across international borders. Therefore, policy planners must consider the long-term impact of their plans such that the harvesting of the Mekong can be conducted sustainably.

### Data availability

Hydrological data are open source and can be downloaded from the MRC data portal at https://portal.mrcmekong.org/home. Precipitation data used in this study is also obtained from the MRC portal.

### Author contribution

**Samuel Chua**: Conceptualization, Data curation, Formal analysis, Methodology, Writing - original draft, Writing - review & editing. **Lu Xi Xi**: Conceptualization, Data curation, Formal analysis, Methodology, Writing - original draft, Writing - review & editing, Funding acquisition. **Chantha Oeurng**: Data Curation, Methodology, Writing - review & editing. **Ty Sok**: Data Curation, Methodology, Writing - review & editing. **Carl Grundy-Warr**: Writing - review & editing.

### Competing interests

The authors declare that they have no conflict of interest.

### Acknowledgements

We want to thank the Mekong River Commission (MRC) for access to the hydrological data. We want to thank the Ministry of Water Resources and Meteorology, Cambodia for their support. This study was supported by the National University of Singapore (R-109-000-227-115; R-109-000-273-112).

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
