# Peer review of "Drastic decline of floodpulse in the Cambodian floodplains (the Mekong River and the Tonle Sap system)"

_Hydrology and Earth System Sciences, 2021_

## Author Comment (AC1)

We thank the reviewer for his/her valuable comments. Here are our responses.

Regarding the research objectives that the Authors want to address, I believe they need to make them more compelling. In particular, some aspects of the objectives have answers even before attempting any analysis. Identifying anthropogenic factors while no information is provided concerning sand-mining, as an example, makes the corresponding results debatable.

We apologise for the misinterpretation here. We gave background information about irrigation and sand-mining in Section 2 because we thought that it would be helpful for readers not familiar with the region to know more about the issues that the region faces. Placing the information at the end of our analyses in the discussion section risks a confusing report for the reader. However, we would be happy to shift them to the back if the Editor also concurs with the reviewer.

We do not understand by what the reviewer means as 'compelling'. Our research covers a critical area of the Mekong basin and highlights observations that have not been previously reported. Beyond listing two anthropogenic responses (which the reviewer disagrees with), our paper also addresses implications for the Tonle Sap Basin and the Vietnamese Mekong Delta as a result of the changes in the flood pulse.

Section 4.1.1, the information and data given in this section can be easily found on the MRC website, so what is the novelty of this section? Additionally, the explanations provided in this section are vague and can be better expressed. For example, lines 174 to 175, where it is "compared to the pre-dam era from 1962-1991, …….." what is observed from figure 3d and e is that the minimum water level seems not to be higher than the minimum water level for pre-dam era for Neak Luong and Chaktomuk stations, at least it is not tangible. I would like to suggest authors revise this part slightly.

After going to the MRC website, we could not easily find the data and information as mentioned by the reviewer and perhaps the reviewer can provide us with the URL link.

On the MRC website, the latest basin report was in 2019 (MRC, 2019). Even so, the report only used data from 2000-2018. The short study duration cannot represent a historical baseline and cannot differentiate the conditions pre-dam and post-dam. As such, the changes in hydrological parameters within the latest 2010-2019 period as presented in section 4.1.1 is a novel finding. Furthermore, even recent research only used data until 2015 (Binh et al., 2020; Li et al., 2017; Yun et al., 2020) and cannot account for the rapid water infrastructure development post-2010 such as the operationalisation of the Nuozhadu Dam in China.

There is no error in our description. We conducted a Welch's t-test to compare the minimum water level values and found a statistically significant ($p<0.05$) increase of +0.10m at Neak Luong. This data is already presented in the supplement. It would be erroneous to judge the significance of change simply by visual analysis of graphs as what the reviewer suggested.

Additionally, no performance index, i.e., $R^2$, is provided for Neak Luong's rating curve to see how reliable this rating curve is. It seems the authors extracted the data presented in Fig. II-3-4 at MRC et al. (2004) to develop these rating curves. Although insignificant, this per se can cause errors.

The reviewer assumes that we manipulated data from the graph below:

[Figure]

The allegations are totally untrue and the reviewer should not make such baseless comments.

The R-square value for the said rating curve is 0.979 and is presented in the same section (MRC et al., 2004).

In section 4.2.1, the authors discussed the water exchange at the TSL in various timeframes while they are different from those defined in lines 121 to 132. Please justify.

We apologise for the confusion. The timeframes listed from lines 121 to 132 are indeed phases by which data was divided. In section 4.2.1, the water exchange at the TSL is listed in these divisions: 1962-1972, 1995-2009 and 2010-2018. The years within the three divisions are still subsets within the original phases determined in lines 121 to 132. Therefore, the timeframes used for the TSL exchange are valid.

Additionally, the average start date has been used to compare mega-dam and pre-dam eras; how do the authors justify this, while, according to the MRC, even a few days would cause a relatively significant change in accumulated flow during water exchange, let alone average date; please clarify

We would like to ask the reviewer to provide the exact reference from MRC for this piece of information. We hope that the reviewer can provide the proper source and reference for his/her arguments.

In Figure 4a, the authors presented discharge data for Prek Kdam hydrological station for various time frames as continuous data, while, according to the MRC data portal, there are significant gaps for some of these years (1964-1972 and 2011 to 2018). Would you please explain how the authors get the missing discharge data since what is observed from this figure differs from the existing data?

We obtained Prek Kdam discharge data primarily from the MRC data portal. We then cross-check gaps in the data records with records from the **Ministry of Water Resources and Meteorology, Cambodia.**

It is not clear what the authors refer to in line 221. According to water level data, the reduction mentioned in this part is not equal year to year. Is what the authors are referring to the "average reduction" in water level during these periods? Presenting a value/percentage for each timeframe seems not interesting as the anthropogenic and natural impacts are monitored year by year.

Please define what is 'interesting'. Presenting year-on-year data might be interesting to the reviewer, but for others, it is interesting to understand the overall hydrological trend that Tonle Sap Lake has experienced from 1996-2019.

In lines 230 to 235, the results are compared with the published research on maximum and minimum water levels influencing the inundated area; however, those references have addressed the inundation changes as a pattern throughout the investigated periods. Also, what is seen from those references in line 231, is that various periods have been determined, which is different from the present work; how these contradicted results are justified, also a compelling discussion is required here.

Ji et al. (2018) found that both maximum and minimum water levels have decreased from 2000-2014. Lin and Qi (2017) found that both maximum and minimum water levels have decreased from 2001-2015. Wang et al. (2020) found that both maximum and minimum water levels have decreased from 2000-2018. We found that both maximum and minimum water levels have decreased from 1996-2019. While slight differences in study periods exist, the trend reported is clear throughout all 3 papers. We do not understand the reviewer's claim of contradiction.

Additionally, the reason given for the differences in results with hydrodynamic models is also questionable. How do authors justify it when the mentioned dams have not been operated yet and the research period is somewhat different? For the explanations given in lines 238 to 241, I would also suggest investigating the flood duration year by year over each period and make a comparison between timeframes.

We wrote 'For instance, reservoirs are being developed at Stung Sreng and Stung Pursat —two key tributaries of the Lake (ADB, 2019a)' (line 235) This means that the reservoirs at Stung Sreng and Stung Pursat only represent two out of many, not two solely. Even so, construction of water infrastructures at both Stung Pursat and Stung Sreng started in 2011, which would have influenced the hydrology at Tonle Sap Lake during our study period of 1996-2019.

We do not question the validity of the hydrodynamic models presented by the models (Arias et al., 2012, 2014; Kummu and Sarkkula, 2008; Piman et al., 2013). We only wish to present a possible reason why the models have not accurately predicted the changes in the Tonle Sap Lake within the past decade.

While a year-by-year analysis would be interesting for the reviewer, it is unfortunately out of the current scope of the paper. If the reviewer is interested, we may suggest an excellent recent paper by Chen et al. (2021).

For section 5.1, I would also recommend using a more precise definition of variables when a timeframe is addressed. For example, one value is given for the pre-dam era, which seems to refer to the average discharge. Could the authors please confirm and readjust accordingly throughout the manuscript? Additionally, the time series discharge data is available for Stung Treng and Kratie stations for the investigated timeframes, while for other ones, discharge data exist for some years, and the authors have tried to obtain the missing data employing rating curves. Would you please explain how the discharge data were obtained for the mega-dam era for Kompong Cham station as no rating curve has been presented for this station? It seems to the reviewer that the missing discharge data for Chaktomuk and Neak Luong stations have been obtained using two rating curves given in lines 111 and 112, and please justify the reliability of these rating curves to see the accuracy of the results presented in Figure 5. According to the MRC data portal, discharge data exist for limited years for Prek Kdam, so please clarify how data were obtained for this station.

Apologies. the data were calculated from the average of wet-season discharges from June-September in the respective eras.

At Kompong Cham the following rating curve adapted from MRC et al. (2004) was used to calibrate discharge records from 2012-2019:

Kompong Cham (KC): $Q_{KC} = (8.869H_{KC} + 29.811)^2(H_{KC} - H_{CC})^{0.3}$ (R²=0.998)

We apologies for our mistake and would include this information in our subsequent revisions.

The same concerns regarding lack of compelling discussion and relatively poor data analysis exist for the remaining parts.

We are deeply disappointed that the reviewer did not finish our manuscript based on his/her presumptive judgement that our data analysis was poor. It seems that s/he has incorrectly assumed that we manipulated our data and thenceforth, dismissed the manuscript outright. The reviewer's unsubstantiated criticism of our 'lack of compelling discussion and relatively poor data analysis' also makes it challenging for us to improve our work further.

For the equations presented in section 4.2.1, please provide the performance index, i.e., R², same as Kummu et al., 2014.

We thank the reviewer for the suggestion, we will include it in the final revision.

I recommend authors polish the paper as there are many grammatical errors. Some are listed below:

While some edits are helpful, we regret to inform that some of the suggested grammatical changes are in fact, grammatically wrong.

line 30, where it is "that also includes the Cambodian floodplains and the Tonle Sap system," it seems that a pronoun problem exists here. Consider removing it.

The sentence is grammatically correct.

Where it is "floodpulse" throughout the manuscript, please separate the floodpulse into two separate words as flood pulse.

We thank the reviewer for the suggestion.

In lines 34, 44, 58, 84, please correct the grammatical errors related to the used verbs and pronouns.

The sentences in lines 44 and 84 are grammatically correct.

I recommend authors use "past tense" for sentences in the "material and methods section."

We thank the reviewer for the suggestion

Line 127, where it is "with a total storage capacity no less than," the preposition is missing.

The sentence is grammatically correct.

Line 154, where it is " The parameters to characterise reverse flow (RF) where water flow from the Mekong to Tonle Sap Lake ….. , it should be "water flows from …..". Same gramtical error is seen in lines 158 and 159.

The sentence is grammatically correct.

Line 232, it appears that that may be unnecessary in this sentence. Consider removing it.

We thank the reviewer for the suggestion.

References:

Arias, M. E., Cochrane, T. A., Piman, T., Kummu, M., Caruso, B. S. and Killeen, T. J.: Quantifying changes in flooding and habitats in the Tonle Sap Lake (Cambodia) caused by water infrastructure development and climate change in the Mekong Basin, J. Environ. Manage., 112, 53–66, doi:10.1016/j.jenvman.2012.07.003, 2012.

Arias, M. E., Piman, T., Lauri, H., Cochrane, T. A. and Kummu, M.: Dams on Mekong tributaries as significant contributors of hydrological alterations to the Tonle Sap Floodplain in Cambodia, Hydrol. Earth Syst. Sci., 18(12), 5303–5315, doi:10.5194/hess-18-5303-2014, 2014.

Asian Development Bank: Irrigated Agriculture Improvement Project: Report and Recommendation of the President., 2019.

Binh, D. Van, Kantoush, S. A., Saber, M., Mai, N. P., Maskey, S., Phong, D. T. and Sumi, T.: Long-term alterations of flow regimes of the Mekong River and adaptation strategies for the Vietnamese Mekong Delta, J. Hydrol. Reg. Stud., 32(October), 100742, doi:10.1016/j.ejrh.2020.100742, 2020.

Chen, A., Liu, J., Kummu, M., Varis, O., Tang, Q., Mao, G., Wang, J. and Chen, D.: Multidecadal variability of the Tonle Sap Lake flood pulse regime, Hydrol. Process., doi:10.1002/hyp.14327, 2021.

Ji, X., Li, Y., Luo, X. and He, D.: Changes in the Lake Area of Tonle Sap: Possible Linkage to Runoff Alterations in the Lancang River?, Remote Sens., 10(6), 866, doi:10.3390/rs10060866, 2018.

Kummu, M. and Sarkkula, J.: Impact of the Mekong River Flow Alteration on the Tonle Sap Flood Pulse, Ambio, 37(3), 185–192, doi:10.2307/25547881, 2008.

Li, D., Long, D., Zhao, J., Lu, H. and Hong, Y.: Observed changes in flow regimes in the Mekong River basin, J. Hydrol., 551(June), 217–232, doi:10.1016/j.jhydrol.2017.05.061, 2017.

Lin, Z. and Qi, J.: Hydro-dam – A nature-based solution or an ecological problem: The fate of the Tonlé Sap Lake, Environ. Res., 158(June), 24–32, doi:10.1016/j.envres.2017.05.016, 2017.

Mekong River Commission, Japan International Cooperation Agency, CTI Engineering International Co., L. and Nippon Koei Co., L.: The Study on Hydro-meteorological Monitoring for Water Quantity Rules in Mekong River Basin., 2004.

MRC: State of the Basin Report 2018, Vientiane., 2019.

Piman, T., Lennaerts, T. and Southalack, P.: Assessment of hydrological changes in the lower mekong basin from basin-wide development scenarios, Hydrol. Process., 27(15), 2115–2125, doi:10.1002/hyp.9764, 2013.

Wang, Y., Feng, L., Liu, J., Hou, X. and Chen, D.: Changes of inundation area and water turbidity of Tonle Sap Lake: responses to climate changes or upstream dam construction?, Environ. Res. Lett., 15(9), 0940a1, doi:10.1088/1748-9326/abac79, 2020.

Yun, X., Tang, Q., Wang, J., Liu, X., Zhang, Y., Lu, H., Wang, Y., Zhang, L. and Chen, D.: Impacts of climate change and reservoir operation on streamflow and flood characteristics in the Lancang-Mekong River Basin, J. Hydrol., 590(September), 125472, doi:10.1016/j.jhydrol.2020.125472, 2020.

---

## Author Comment (AC2)

We thank the reviewer for his/her valuable comments. We have looked through the suggestions and will make the necessary amendments.

The first issue is that the authors did not consider the changes in climate in the basin, especially precipitation. The Mekong River Basin has experienced large inter-annual and -decadal variations in precipitation. Without considering the changes in precipitation, which makes it very hard to draw the conclusions.

Indeed in our manuscript, we recognise the significance of precipitation and have considered its impact from lines 290-295 and Figure 8 as follows:

*Also, as observed in Figure 8, measured rainfall in the Cambodian floodplains has remained roughly constant from 1960-2019, in line with observations via other sensing methods (Raghavan et al., 2018; Singh and Qin, 2020; Thoeun, 2015). Thus, the observed reduction of flood discharge in the Cambodian floodplains cannot be attributed solely to either upstream developments or natural climatic variability – local anthropogenic factors are likely the main reason.*

[Figure]

***Figure 8. Measured precipitation data at Kompong Cham and Kratie from 1960-2019. At both stations, there are no statistically significant trends in rainfall within the data period.***

Second, the authors point out the increasing sand-mining activity within the river channels, especially in Cambodia (from Kompong Cham to the Vietnamese border); and if such sand-mining has altered the channel morphology, which can lead to change in channel hydrology and observations. Hence, without taking such impacts, using the water level data from the stations within the river channels would not be reliable.

The water level data obtained for the respective stations in the Cambodian floodplains are actual measured data. Our analysis of floodpulse characteristics is based on actual values and is thus, reliable. Furthermore, regardless of its cause (changes in water amount or changes in channel morphology), the fact remains that water levels have generally decreased in the flooding season, and with it, a reduction of the floodpulse inthe floodplains along the channel.

Third, the authors showed the water level reduction due to incision and water withdrawal (in Table 3), however, there is very limited description of the method, which made it hard to judge the results.

Thank you for the suggestion. We will include a more detailed description of the method as per below:

*(Line 325)*

*We estimate the contribution of sediment decline and irrigation to the reduction of water levels by comparing water level changes at the start and end of the flood season. At the start of the flood season in July, minimal water will be diverted for irrigation as the fields will be flooded naturally. Thus, changes in water levels in July will be caused predominantly by incision. Conversely, at the end of the flood season in December, water from the receding floodwaters will be diverted for irrigation in anticipation of the dry season. It is also during this period that diversion amount is at its greatest. Therefore, the changes in water levels in December will reflect influences from both irrigation and incision.*

*In the absence of man-made hydrological alterations, water levels at upstream Kratie will be tightly coupled with those at downstream stations throughout the year. However, as seen in Figure 9, water levels at Neak Luong and Chaktomuk have been decreasing with respect to water levels at Kratie. In other words, for the same water level at Kratie, the water level at Neak Luong/Chaktomuk has decreased from the 1960s to 2010s.*

*To estimate the separate contribution of incision versus water withdrawal as seen in Table 3, we compared water level changes ($\Delta WL$) in July and December. Since water level reduction in July in caused mainly by incision, it follows that:*

$$\Delta WL_{caused\ by\ incision} = \Delta WL_{July}$$

*In December, water level changes are caused by a sum of both incision and water withdrawal. Thus,*

$$\Delta WL_{caused\ by\ water\ withdrawal} = \Delta WL_{December} - \Delta WL_{caused\ by\ incision}$$

$$\rightarrow \Delta WL_{caused\ by\ water\ withdrawal} = \Delta WL_{December} - \Delta WL_{July}$$

*However, this method assumes that channel morphology at Kratie has remained constant from 1992-2019. If the channel at Kratie have undergone erosion, then our estimates for water level reduction due to both incision and water withdrawal would be lower than actual values.*

In section 4.1.3 changes to rise and fall rates: the authors discuss the changes to rise and fall rates in different drought periods, and found increases in these two indicators. They argue that the increases hint at anthropogenic hydrological regulation in the region. However, I don't see a clear connection between the indicators and anthropogenic hydrological regulation. Please elaborate more.

Changes in rise rates and fall rates reflect influences of upstream water infrastructures. During the rising limb of the wet season, reservoirs would have to release the water stored during the dry season in preparation for the incoming water (Richter et al., 1997; Singer, 2007). Also, the presence of irrigation canals increases the conveyance speed of floodwaters across the floodplains, resulting in an increased rise rate. After the wet season, upstream reservoirs or irrigated fields would retain water (Cochrane et al., 2014). As flows to the main channel is reduced, the fall rate would be correspondingly higher.

In Figure 4: The authors showed the changes in discharge in Prek Kdam and water level in Kompong Luang in different time periods. However, the time periods are different, which are not comparable directly. Would be better to show the figure with the same time periods, e.g., 1996-2009, 2010-2018.

Thank you for your suggestion. We have altered our calculations and have made the necessary changes.

[Figure]

References:

Cochrane, T. A., Arias, M. E. and Piman, T.: Historical impact of water infrastructure on water levels of the Mekong River and the Tonle Sap system, Hydrol. Earth Syst. Sci., 18(11), 4529–4541, doi:10.5194/hess-18-4529-2014, 2014.

Richter, B., Baumgartner, J., Wigington, R. and Braun, D.: How much water does a river need?, Freshw. Biol., 37(1), 231–249, doi:10.1046/j.1365-2427.1997.00153.x, 1997.

Singer, M. B.: The influence of major dams on hydrology through the drainage network of the Sacramento River basin, California, River Res. Appl., 23(1), 55–72, doi:10.1002/rra.968, 2007.

---

## Author Response (AR1)

Dear Editor,

Thank you for your valuable suggestions. We have deeply revised major aspects of our paper following the comments by the editor and reviewers. The revised portions are in red. Generally, we have consolidated our arguments, clarified our methods and reorganised some sections. Also, we have taken the opportunity to update several sections with recently published literature on the Cambodian floodplains. We hope that the revised version is now much more clearer to readers.

1) The major finding of the paper is local (Cambodian) anthropogenic factors are likely the main reason for the drastic decline of floodpulse. But the paper introduces the problem from upstream dams and the three study periods were divided based on upstream dam construction. This would be very misleading. When the editor read the paper, I was always looking for your evidence of impacts from upstream dams on the floodpulse because the term "mega-dam period" always reminded me to do so. Furthermore, as the local anthropogenic factors are likely the main reason, so the authors should introduce the local anthropogenic activities in more detail. I understand the investigation data may be very rare, the descriptive materials are still helpful. Also, the study period division should also consider both upstream and local factors. The period division is very important for attributing studies, a lot of studies adopted trend and abrupt changing point analysis methods. So more explanation is required for the period division.

Thank you for the suggestion. We have included additional information on anthropogenic activities that we hope can enhance readers' knowledge of the area. While our paper argues that local factors are likely the main reason, we must still view the hydro-geomorphological changes within the context of the wider Mekong basin. Thus, we thought the separation of the study period by the three eras can allow readers to view the local changes vis-à-vis upstream changes.

Heading your advice, we have included additional justification and information on how upstream and local factors can be viewed in tandem (Section 3.2). In the discussion, we further elaborated on the competing influences of the upstream dams (Section 5.1) versus that of local factors (Section 5.2 and 5.3). Hopefully, this separation allows the reader to better appreciate the various drivers of hydrological changes in the Cambodian floodplains.

2) Data quality is of a big concern in the Mekong studies. The Referees' comments also elaborate this issue. The authors should find some way to demonstrate it. For example, Figure 5 presents the wet season discharge on the Cambodian floodplains during the two study periods, which is very useful and indicative. Is that possible to show the annual discharge as well and use them to conduct a water balance analysis to verify the data quality? Water inflow and outflow should be balanced at a longer time scale. Or the authors can utilize the lake water storage change results at the annual scale if possible.

We concur that our previous documentation of data sources was confusing. Thus, we have included additional information in 3.1 on how future research can obtain the same data to test our findings. At several stations, we used published rating curves from MRC et al. (2004) to estimate discharge values. Following Reviewer 2's concerns about the quality of the curves, we tested the accuracy of the curves and compared them to actual discharge as documented in Section 3.1. We found that the accuracy is

generally high among all stations ($R^2$ >0.99). Thus, subsequent analyses using the predicted discharge data should also be accurate.

However, it is not possible to do a water balance analysis as the editor suggested due to insufficient data. For example, we do not have physical information on the exact quantity of overland flow across the floodplains.

3) Data sharing: As the authors collated a lot of data from MRC and other sources and make efforts to clean them. Is that possible to share the collated data in some way (so the followers can easily replicate the results and go beyond)?

While we will be happy to share our data with any future works, the data that we have obtained from MRC are subjected to their data use licences and copyright. Nonetheless, future authors can register with MRC and obtain the most updated data directly from them. Non-csv data can also be accessed directly in-browser on the MRC data portal at https://portal.mrcmekong.org/home..

Minor comments:
1) The abstract can be more conclusive. The current version contains quite a few numbers but lose the informative conclusion. Also, the main reason for decline of floodpulse is not clearly stated.

We have rewritten the abstract and conclusion and emphasized our main findings: that upstream contributors are not the main reason for the decline of floodpulse, and that local factors should also be considered.

2) Figure 2: is that a typical annual water level / discharge figure or a virtual one? Please check.

Thank you for pointing this out. This is a virtual figure and we have updated the graph label accordingly.

3) P10L184, the authors claim more area of Cambodian floodplains are now permanently inundated during the dry season. But as I can see from the Figure 3, the dry season water level is well below flood threshold (dashed line). Why the authors say "permanently inundated" when water level is not higher than flood threshold. Please explain more.

Apologies, we meant that the riverbanks are more permanently inundated. We have edited to as follows: "…more areas of the riverbanks are now permanently inundated during the dry season"

4) P17L311, this paragraph is confusing. I cannot understand why the authors mention running dry canals when talk about the impacts of water regulation. More meaning discussion should be the storage capacity of reservoirs, the area of expanded paddy field, the water demand and irrigation amount for these expanded paddy field, etc.

We agree that our mention of water wastage due to poorly designed infrastructure lacks more information and might be misleading. Thus, we have removed the paragraph as it is irrelevant to our central thesis.

5) P18L328, the authors state that water from the receding floodwaters will be diverted for irrigation in anticipation of the dry season. This statement is more like an assumption. Any evidence to support it?

We have provided additional information in Section 5.2 about the planting calendar in Cambodia. We drew upon various sources that describes the cropping and irrigation cycles in the Cambodian floodplains (Cramb et al., 2020; MRC, 2009; Phengphaengsy and Okudaira, 2008).

6) P20L373, the authors concluded that the Tonle Sap Lake has released about 6.2 km$^3$ more water annually to the Mekong during 2010-2019 as compared to 1962-1972. Is that possible to find the data (e.g., GRACE) to validate these results?

To confirm our result that the Tonle Sap Lake has released more water, we used reconstructed water level data by Guan and Zheng (2021) to predict the Lake volume from 1960-1990. Compared to current lake volumes, we found that water volume has indeed decreased. Therefore, our claim that the Tonle Sap Lake is losing more water to the Mekong is valid. We have also edited 5.4 to include this justification.

References:
Cramb, R., Sareth, C. and Vuthy, T.: The Commercialisation of Rice Farming in Cambodia, in White Gold: The Commercialisation of Rice Farming in the Lower Mekong Basin, pp. 227–245, Springer Singapore, Singapore., 2020.

Guan, Y. and Zheng, F.: Alterations in the Water-Level Regime of Tonle Sap Lake, J. Hydrol. Eng., 26(1), 05020045, doi:10.1061/(ASCE)HE.1943-5584.0002013, 2021.

Mekong River Commission, Japan International Cooperation Agency, CTI Engineering International Co., L. and Nippon Koei Co., L.: The Study on Hydro-meteorological Monitoring for Water Quantity Rules in Mekong River Basin., 2004.

MRC: Regional Irrigation Sector Review for Joint Basin Planning Process., 2009.
Phengphaengsy, F. and Okudaira, H.: Assessment of irrigation efficiencies and water productivity in paddy fields in the lower Mekong River Basin, Paddy Water Environ., 6(1), 105–114, doi:10.1007/s10333-008-0108-z, 2008.

---

## Editor Decision (ED1)

Dear authors,

I reviewed the paper, the Referees' comments and your responses carefully. I believe the Referees' comments can stimulate more depth thinking and will ultimately improve the quality of your work substantially. I generally agree with the comments and believe the paper need a major revision before its potential publications on HESS.

In addition to the Referees' comments, I have my own comments as follows.

General comments: the issue of drastic decline of floodpulse in the Cambodian floodplains has received a lot of attention due to its potential great impact on the ecosystem, fishery, livelihood in the Mekong floodplain and delta regions. This paper collated a large amount of data to quantify such decline and tries to figure out its driving factors. The findings are of significance for proper water resources management and transboundary cooperation. However, the quantification and attribution are complex for such a large scale and data shortage region. The following points should be addressed to consolidate the conclusions:

1) The major finding of the paper is local (Cambodian) anthropogenic factors are likely the main reason for the drastic decline of floodpulse. But the paper introduces the problem from upstream dams and the three study periods were divided based on upstream dam construction. This would be very misleading. When the editor read the paper, I was always looking for your evidence of impacts from upstream dams on the floodpulse because the term "mega-dam period" always reminded me to do so. Furthermore, as the local anthropogenic factors are likely the main reason, so the authors should introduce the local anthropogenic activities in more detail. I understand the investigation data may be very rare, the descriptive materials are still helpful. Also, the study period division should also consider both upstream and local factors. The period division is very important for attributing studies, a lot of studies adopted trend and abrupt changing point analysis methods. So more explanation is required for the period division.

2) Data quality is of a big concern in the Mekong studies. The Referees' comments also elaborate this issue. The authors should find some way to demonstrate it. For example, Figure 5 presents the wet season discharge on the Cambodian floodplains during the two study periods, which is very useful and indicative. Is that possible to show the annual discharge as well and use them to conduct a water balance analysis to verify the data quality? Water inflow and outflow should be balanced at a longer time scale. Or the authors can utilize the lake water storage change results at the annual scale if possible.

3) Data sharing: As the authors collated a lot of data from MRC and other sources and make efforts to clean them. Is that possible to share the collated data in some way (so the followers can easily replicate the results and go beyond)?

Minor comments:
1) The abstract can be more conclusive. The current version contains quite a few numbers but lose the informative conclusion. Also, the main reason for decline of floodpulse is not clearly stated.
2) Figure 2: is that a typical annual water level / discharge figure or a virtual one? Please check.
3) P10L184, the authors claim more area of Cambodian floodplains are now permanently inundated during the dry season. But as I can see from the Figure 3, the dry season water level is well below flood threshold (dashed line). Why the authors say "permanently inundated" when water level is not higher

than flood threshold. Please explain more.

4) P17L311, this paragraph is confusing. I cannot understand why the authors mention running dry canals when talk about the impacts of water regulation. More meaning discussion should be the storage capacity of reservoirs, the area of expanded paddy field, the water demand and irrigation amount for these expanded paddy field, etc.

5) P18L328, the authors state that water from the receding floodwaters will be diverted for irrigation in anticipation of the dry season. This statement is more like an assumption. Any evidence to support it?

6) P20L373, the authors concluded that the Tonle Sap Lake has released about 6.2 km$^3$ more water annually to the Mekong during 2010-2019 as compared to 1962-1972. Is that possible to find the data (e.g., GRACE) to validate these results?

Hope all the comments can help you improve the manuscript. Please be noted, the revised manuscript will be subject to another round of reviewing, which will not guarantee its final publication on HESS.

Kind regards,
Fuqiang

---

## Author Response (AR2)

We thank the reviewer for his/her comments and have responded to them accordingly. The revised portions are in red. We hope that the revised version is now much clearer to the reader.

1. This is an important transboundary issue in Eastern and Southeastern Asia that is suited for the special issue. However, I do not recommend the authors publishing the paper in the present form. The hydrological analysis, while likely time consuming, lacks hydrological rigour and ignores the findings of much past work showing that new insights of this complex system can only be gained by greater consideration of the 3 dimensional hydrological balance (e.g., Kummu et al., 2014), which was recently reiterated at by Kallio and Kummu (2021) in pointing out the limitations of the recent Wang et al (2020) analysis (and the reply of Wang et al (2021).

In Kallio and Kummu (2021), the authors compared modelled discharge (without dam operations) to actual observed discharge at Stung Treng and found an increasing trend during the dry season and a decreasing trend during the wet season. Thereafter, the authors concluded that anthropogenic changes are a better indicator of the Tonle Sap Lake (TSL) hydrology as compared to precipitation changes.

The subsequent reply by Wang et al. (2021) challenges the aforementioned conclusion. Specifically, they pointed out that the observed discharge at Stung Treng were much larger than that at Chaing Saen. Also, the water level at Kratie showed greater correlation with the inundation of TSL than at any other stations along the Mekong. These evidences presented showed that it is unlikely that Chinese dams are fully responsible for changes in the TSL.

Indeed, these studies have shed light on the complexity of the hydrology of TSL and we have acknowledged their findings and contributions to literature accordingly. Nonetheless, there are differences between these papers and our work here.

- These papers only considered hydrology at the TSL and have not considered the influence of the surrounding Cambodian floodplains. Given the close connection between the TSL and the Cambodian floodplains, it is important to consider the broader region as a whole. Focusing only on the TSL might lead to missing out of key elements in the local geography – that local factors in the form of irrigation and channel incision are important factors affecting the floodpulse in the region.

- In a way, our study imparts an additional layer of insight onto these existing works. For example, we showed through an alternative method that water infrastructures (dams, irrigation diversion etc.) upstream of Stung Treng are unlikely to be the main agent of hydrological change (Section 5.1), further extending the arguments by Wang et al. (2021) that the impacts of Chinese dams within the region is over-estimated. Furthermore, we considered hydrological records from 8 stations, more than that in Kallio and Kummu (2021) or Wang et

al. (2021). **Thus, our methods can be viewed as empirical tests to the model-driven approaches of the two papers.**

- **We stress that our objective was never to construct a hydrological balance.** To do so would require data on evapotranspiration, groundwater movement and overland flow – data that is almost impossible to obtain. **Instead, our research is solely focussed upon the quantification of floodpulse along the Cambodian floodplains using 60 years of data from 1960-2019**. Creating a hydrological balance with the dearth of data would require formulation of numerous assumptions, which would challenge the validity of the balance itself.

2. The study considered is arguably limited too, as it only considers very few spatial measurements of water depth and (sometimes potentially flawed) discharge. Further it includes rainfall from only locations. The calculations based on this limited set of information likely do not allow a full interpretation of the myriad flow processes affecting this complicated hydrological setting. In the end, the main findings are basically the same from those published by Kummu et al (2014) that uncertainty in the hydrology of the tributaries of the Tonle Sap largely prevent closing the water balance of this complex system. The other aspect preventing closure of the budget is uncertainty in the groundwater dynamics.

In our study, we considered over 60 years of water levels and discharge records across 8 different stations within the Cambodian floodplains. We hope that the reviewer can clarify what s/he means by "potentially flawed discharge". For our precipitation data, we obtained observed readings from Kompong Cham and Chaktomuk which are stations located centrally within the Cambodian floodplains. As the records are actual physical measurements, the readings are arguably not less reliable than model-based approaches such as TerraClimate or GMFD that might defer from ground truths.

Kummu et al. (2014) found that 53.5% of water in the TSL originates from the Mekong, 34% from its tributaries and 12.5% from precipitation. **Our main findings – that the observed decline in floodpulse in the Cambodian floodplain is caused mainly by local anthropogenic factors – had not been mentioned within Kummu et al. (2014) at all.** Straightaway, our geographical coverage of the entire Cambodian floodplains from Stung Treng to Neak Luong is much more extensive than the study area of Kummu et al. (2014) – just the TSL. **Therefore, our research is novel and not a repetition of Kummu et al. (2014) as what the reviewer suggests.** We reiterate that it was never our aim to close the hydrological budget and this is out of the scope of our research paper.

3. Further, the findings do not go farther than echoing the findings in the most recent works (Wang et al., 2020; 2021; Kallio an Kummu, 2021, Ng and Park, 2021). Thus, the bulk of the

conclusions are largely the information that is already alluded to in the introduction (or should be alluded to with careful inclusion of the 2021 papers). I am struggling therefore to see what the main contribution of this new analysis is other than "contributing to the discussion", but in a quality that is not quite at the level expected for HESS. Better would be if the authors invested much effort in addressing the theme of the special issue by using their "estimates" to re-enforce the issue of the complexity in determining the lake water balance and possible drivers, which are both local and transboundary in nature.

Ng and Park (2021) argues that sand-mining at Phnom Penn have caused water levels at Phnom Penn Port to decrease, and subsequently caused the TSL to shrink. Indeed, the four papers listed by the reviewer contributes greatly to discussion on the TSL. However, their focus on the TSL means that the hydrology of the larger surrounding Cambodian floodplains has been neglected. Thus, the main difference between these works and our study is that we have considered the hydrology and various anthropogenic drivers across the broader Cambodian floodplains, not just the TSL as these papers have done.

To reiterate our arguments, our study has several novel contributions:

- We quantified the decrease in floodpulse within the Cambodian floodplains (from Kratie to Neak Luong) in terms of flood parameters
- We identified from actual observed records that the reverse flow along Tonle Sap River has been decreasing from 1962-2018.
- We argue that these changes cannot be fully explained by upstream water infrastructures – local drivers of irrigation and channel incision are also important factors to be considered.

**We are confident that we have not simply copied the conclusions of the works that the reviewer cited. We use a very holistic dataset and our findings are novel and of interest to the scientific community.**

We disagree with the reviewer's recommendation to "determine the lake water balance" because, as we stated earlier, this is not the scope of our research topic. Again, we must stress that we are not aiming to craft a hydrological budget for the TSL as what the reviewer has suggested.

4. Also, to be publishable in this journal, much greater care is needed in telling the comprehensive story, as well as addressing the limitations of the calculations at hand--and perhaps the uncertainties in those of other studies. Currently, the message that the reader is left with is that too much attention has been focused on the role of upstream dams in the past, but again, even this focused topic is not addressed in a meaningful discourse that aligns with the specific theme of the special issue. Largely, incomplete explanations of processes related

to the other important drivers are given (e.g, how exactly downstream sand mining affects upstream hydrology).

We disagree with the reviewer that we have focussed "too much attention … on the role of upstream dams in the past". Even though our paper argues that local factors are important drivers of hydrological changes, **we must view these changes within the context of the wider Mekong Basin.** Therefore, upstream dams should be part of the discourse given its capabilities to regulate flow downstream. **A discussion without considering the impacts of upstream water infrastructure would be missing the elephant in the room – arguments on local impacts must be compared against the competing influences of upstream factors**.

We are unsure of what the reviewer mean by "incomplete explanation". The three anthropogenic factors that we investigated – upstream dams, water withdrawal and channel incision – affects the downstream hydrology of the Cambodian floodplains. Their relationships with respect to each other are not within the scope of this paper. Like the reviewer, we are also unsure how downstream sand-mining at Neak Luong can possibly affect dam operations in China.

5. Finally, I have questions regarding some of the methods, which are addressed in the points below. Importantly, one question relates to uncorrecting the corrected stream rating curve at a critical location that informs on the changes in discharge to the area in question, and on the role of dams on reducing flows. In practice one adjusts a rating curve when the old one is no longer valid. By adopting the old curve (ignoring the new curve) for the new calculations in this paper, one wonders if the authors are corrupting their calculations. Even if explained elsewhere, sufficient details are needed here for the reader to have confidence in the calculations. Nevertheless, this issue of measurement uncertainty relates to prior calls (e.g., Kummu et al 2014 and likely others) for better and more comprehensive measurements of hydrological phenomena needed to study the Tonle Sap water balance with accuracy. In conclusion, I am hoping this is a very rough draft submitted hurriedly to make the initial deadline of the special issue and that the authors are already undergoing a much more comprehensive assessment to provide an engaging, objective story regarding the issue of Tonle Sap, which is increasingly tragic.

We apologise for the incomplete documentation of our methods and will further clarify our methods in the later sections. Like the reviewer (and many within the scientific community), we support the call for better hydrological data coverage for the Mekong.

While we thank the reviewer for his/her valuable comments, we would like to reiterate that our study is not focussed on only the TSL, as the reviewer has suggested. Our study area is the entire Cambodian floodplains from Stung Treng to Neak Luong. Tonle Sap Lake (and Tonle Sap River)

only represents part of the floodplain. Therefore, there is much more novel insights to be gleaned from our study as compared to other papers that only focus on the TSL.

6. The introduction seems a bit dated, not really discussing the issue based totally on what is known (or debated): a) Mekong flows are reduced and certainly have an effect on the the water levels in Tonle Sap Lake; (b) Climate has had some influence on the hydrology of the entire region (especially the recent "dry" conditions in the region); (c) intense sand mining downstream of the lake is likely a culprit in changing the flow regime of the Mekong, potentially contributing to lake level changes; (d) anthropogenic changes in tributaries above the lake affect inflow to the lake (e) dams and diversion on other mid and lower stream tributaries of the Mekong affect flows in the river as well as other rivers on the Cambodian floodplain; and (f) and agriculture intensification in the floodplains may also affect flows. If these things are "known" how can they be conclusions to the paper? What is not known are the contributions and their combined effects. Importantly, the authors need to emphasize the novelty of their findings.

We structured our introduction with the aim of setting the stage for scholars unfamiliar with the region. We agree that the issues raised by the reviewer has been known and debated by other scholars. However, full coverage of all of these issues and their impacts of the Mekong basin is out of the scope of the paper. Literature reviews such as Hecht et al. (2019); Pokhrel et al. (2018) have reviewed some of these issues and their consequent impacts.

Within the scope of our paper – the Cambodian floodplains – we recognised the various competing drivers of upstream dams, water withdrawal and channel incision and have provided additional information in Section 2 and 5.

Our key areas of novelty are presented in Line 53 to 57:

"we offer novelty in two ways. First, we studied the Cambodian floodplains in its entirety, as compared to other authors who only investigated the Tonle Sap system (Chen et al., 2021; Kummu and Sarkkula, 2008) or the Mekong system (Binh et al.,2020). Second, we synthesised knowledge of the various anthropogenic drivers in the Cambodian floodplains and associated them with observed hydro-geomorphological impacts. In so doing, we present the implications of current human activities on the Cambodian floodplains and the wider region."

7. I think once the finalized message of the paper is determined, the title and abstract can be tuned to reflect that story. Is it simply the decline in the flood pulse (two words not one) or is there more to it? The decline in the flood pulse is already known.

We thank you for the suggestion. Currently, we feel that the title/ abstract already reflects the nature of our research. Nonetheless, we are happy to change them if the editor also concurs with the reviewer.

To our knowledge, there has not been any studies that **quantified** the decline of floodpulse in the Cambodian floodplains using a more extensive range of observed data as this paper. To the stakeholders of the region, it is insufficient to say that flooding season has decreased. For example, they would want to know **exactly** how many days the flooding season has decreased by. In this aspect, this paper has provided the exact quantity the stakeholders and other scientists can take reference from.

8. The estimate of the "water withdrawal rate" from the floodplain is not believable with the simple, indirect methodology.

We are not sure what the reviewer means by "believable". A simple methodology might not be a bad methodology and vice versa. Again, we want address the water balance is not the main focus of this paper.

9. Lines 35 to 40. The brief inclusion of the studies elsewhere are not needed here as they distract from information that is needed to explain the Tonle Sap issue.

As mentioned, our study encapsulates the wider Cambodian floodplain region, not just the TSL. We feel that inclusion of this paragraph allows our reader to better understand the importance of the floodpulse within the floodplains and wider Mekong basin. Nevertheless, we will remove this section if the editor also concurs with the reviewer that this section is a distraction.

10. Line 46. The spatial (and temporal) extent of the Cambodian floodplain area, above and below the lake, should be defined; and an explanation of the hydrological processes operating on this area is needed (what are the boundary conditions?). Make sure to refer to the map.

Indeed, the delineation of the Cambodian floodplains has been done in Section 2 and the corresponding map in Figure 1. The floodplains are the light green sections as indicated in the legend.

Again, we like to remind that the Cambodian floodplains extend beyond the TSL.

11. Line 48. A map is needed showing the Lancang dams, as well as any other dams and features referred to in the paper. I was unable to follow the story without opening Google Maps.

Thank you for the recommendation. We updated Figure 1 to show the entire Mekong Basin.

12. Line 53: The "surface hydrology" of the floodplain system was studie, but only through flow on two rivers and one lake depth. This is not comprehensive.

Sorry, we cannot find the mention of "surface hydrology" as cited. In fact, we did not use the term in the entire paper. As our intention is not the construction of water budget, our extensive coverage of the study area using surface hydrological data is sufficiently comprehensive.

13. Line 55: The "synthesis" amounts to a cursory description, but lacking support data and critical consideration.

Our discussion of anthropogenic drivers – upstream dams, water withdrawal and channel incision – is built upon data and reasoning (Section 5.1 to Section 5.3). We do not understand what the reviewer meant by "cursory" and "lacking support data and critical consideration".

14. Figure 1 Caption. Please provide more details and descriptions of important information.

Thank you for the recommendation. We have added in more details and descriptions.

15. Figure 1. Where are the areas of intense sandming and irrigation (See Ng and Park, 2021).

We have indicated possible irrigation canals in Figure 1 (grey lines) and referred to them in the map legend.

For sand mining, there is a lack of official documentation on its activities. Therefore, there is much uncertainty on the exact location of sand mining operations. Even though Ng and Park (2021) has published a map of possible mining locations (Figure A), their map do not correspond to those published by Hackney et al. (2021) (Figure B).

[Figure]

Figure A. Location of sand mining as per Ng and Park (2021)

[Figure]

Figure B. Location of sand mining as per Hackney et al. (2021)

For instance, Ng and Park (2021) claimed that mining operations along the Tonle Sap River is minimal while Hackney et al. (2021) asserts that the Tonle Sap River is a hotspot for sand mining. These differing views make it hard to establish a consensus on the exact sand mining spot within the Cambodian floodplains.

16. One issue regarding understandability of the paper is the tendency of using "upstream" in reference to tributaries to the lake, tributaries to the Mekong in the vicinity, locations far above. Please be exact and descriptive and provide reference on maps.

Thank you for your suggestions. We meant "upstream" to be regions of the Mekong above the Cambodian floodplains, aka above Stung Treng. We have provided additional clarification in the main text.

17. Lines 79-82. Where exactly is the Chi River with respect to the lake and what are the known effects (use data)? What is meant near the floodplain and how is that more relevant than being far away when river discharge is considered? Please show the Chi and the other S3 dams on a map. What are the details regarding these dams?

Thank you for your suggestions. We have included their locations in the updated Figure 1. More details on their hydrological impacts can be found in works by other scholars (Arias et al., 2014; Cochrane et al., 2014; Piman et al., 2013).

18. Lines 83-92. It is important to show where this area is in relation to the lake and rivers, as up to 31% of the low season flow of the Bassac and Mekong Rivers could be consumed. How can this estimate simply be glossed over and not explored?

We have already indicated possible irrigation canals in Figure 1 (grey lines) and referred to them in the map legend.

19. Lines 95-100. Check out the new Ng and Park (2021) paper and rewrite accordingly.

Thank you for your suggestion. We have written a citation to Ng and Park (2021) as requested.

20. Section 3.1 Rewrite for clarity.

Thank you for your suggestion. We have rewritten for clarity.

21. Line 110. Regarding rainfall from only 2 stations: How is this representative of a huge area? Others have estimated rainfall for the region using much more data (e.g, Wang et al 2020).

While others have estimated rainfall for the region using much more data, the data are afterall, estimates. For example, the data used in Wang et al. (2020) is derived from GMFD, which in turn is based on TRMM rainfall data. While comprehensive, these data also needs to be validated against ground truth. In this aspect, our rainfall data are entirely obtained from ground stations and thus could be regarded as the most accurate source of data in the region.

The purpose of showing rainfall data is to merely illustrate the rainfall changes have not been extreme during the study period. We are not using the data for modelling or budget purpose. Even if we include additional stations, the trend is still the same as the two stations referenced in our paper- that rainfall patterns have been mostly constant from 1960-2019 (Figure C). We selected Chaktomuk and Kompong Cham stations because of two main reasons. One, they are located centrally within the Cambodian floodplains. Two, that their rainfall records are of better quality than the other stations.

[Figure]

Figure C. Yearly Rainfall at 8 stations located within the Cambodian floodplains.

22. Lines 115. Please check the accuracy of the equations. Also, HESS uses numbering on equations correct? Better check. Why are the R2 values so high? What is the timing of the data (daily, monthly, yearly)?

Thank you for pointing out our mistakes. We have added numbering on our equations. The data are in daily intervals.

23. Lines 124-130. Steung Treng is the primary station used to judge flows as affected by the dams, yet the authors alter the rating curve, which makes the most recent flows higher. Why assume the curve wasn't adjusted by the operators because it was wrong and predicting too high of values for a long time?. This "adjustment" affects the validity of the assessment. THIS IS A MAJOR ISSUE that must be addressed in this issue. I didn't read the prior paper, but discussion is needed here to ensure the reader that this is not egregious data manipulation that just so happens to support the story.

At Stung Treng, a hitherto unreported rating curve was adopted on 1 January 2005, with a sudden drop of 623 cms in discharge reading from 31 December 2004 to 1 January 2005 without the corresponding drop in water levels (Figure D). Without further calibration of discharge readings from 2005 onwards, it is erroneous to compare these readings to prior data.

[Figure]

Figure D. The top graph showed a fundamentally different rating curve being employed at Stung Treng from 2005 onwards. This large change in rating curve cannot be the result of natural processes such as channel change as the timing of the change is too abrupt. The bottom graph further shows how discharge changed without a corresponding change in water level. These evidences point to the presence of potential irregularities in the official records so a calibration was undertaken. Graph source: Lu and Chua (2021).

24. Figure 2 is useful, but all the mentioned locations need to be shown on a map; and here, something is needed to identify what water body they affect and where.

The purpose of Figure 2 is to show that most of the water infrastructure developments (be it dams or irrigation projects) occur during the mega-dam era from 2010 to 2019. We have mapped the location of the dams as seen in Figure 1. However, to quantify exactly the hydrological impact of each mentioned location would be out of the scope of this paper.

25. Section 3.2.1. Give the equations numbers please. Are both THRESHOLD and FT needed? Put the units in "( )" not with "/" to make it more clear to the reader.

Thank you for the suggestions. We have made the necessary changes.

26. Figure 3. Rise rate and fall rate are not accurately displayed. What is shown are the depths. You need to show the slope of the line from the start date to the max (for rise rate) for example (e.g a 45 degree angle on the figure).

Thank you for the suggestions. We have changed Figure 3 accordingly.

27. Regarding STATISTICS. Is the use of "validated" a misprint? Have a read again on statistical inference testing for wordiing. I assume it is meant that the Welch test was used because it doesn't assume equal variances; and it was used to assess if the two samples had statistically different means.

Thank you for pointing out our mistake. We meant "conducted".

28. Line 173. Right after the PARAMETRIC Welch test is mentioned, the NONPARAMETRIC Mann-Whitney test is introduced. Why not simply only use nonparametric tests for everything?

We used parametric Welch test because we want to compare the mean of the various distribution. In contrast, Mann-Whitney test was used only because we cannot determine the "mean" of a date range.

29. Lines 176-177. Equation numbers again, and put the units in (). Actually check what HESS recommends.

Thank you for your suggestion. We have rectified our labels accordingly.

30. Section 3.2.3. I would redo this section very carefully and look at net discharge for more combinations than only 2 months. When I look at the various calculations of discharges and water depths over time I am not sure I see a consistent pattern. I would spend some time on thinking this analysis through carefully. Problematic is that this analysis only considers surface flow in three large channels, but I am guessing water is moving with more complexity. THIS IS A MAJOR ISSUE TO INVESTIGATE FOR ACCURACY.

It is wrong to say we only look at net discharge for "only 2 months". We considered the whole wet season from June to September. We are not sure what the reviewer meant by "consistent pattern". At the annual scale, the fluctuation of discharge across the dry and wet season is expected and common knowledge. What we are trying to do here is to quantify how much water is gained/lost at the Cambodian floodplains.

We think that the reviewer has misunderstood our aims. As mentioned, we are not creating a hydrological budget of the Cambodian floodplains. To do so would require much more data and calculations. Neither are we trying to model surface flow across the Cambodian floodplains. Our objective here is simply to estimate the amount of water loss/gain by considering the difference in discharge at the entrance at the floodplain (Stung Treng) versus that at the exit of the floodplain (Neak Luong/ Chaktomuk).

31. Line 207. Where exactly are the water infrastructure developments located?

The water infrastructure developments are located within the Cambodian floodplains. According to our results, their impacts on hydrology from Kratie to Neak Luong/Chaktomuk indicated their large spatial extent across the region. The locations of possible irrigation canals are shown in Figure 1.

32. Figure 3. Why only show water level and not discharge, if even in the supplement? Does one arrive at a different result if Q is used?

We showed only water level because our flood pulse parameters (Section 3.2.1) are based on water level data. Furthermore, the water level data are actual observations from ground records, ensuring that the data quality is high and reliable. Given the tight coupling between water level and discharge, we would reasonably hypothesise that similar results will be obtained if Q is used. However, when we consider changes to the channel, then the two variables will represent different meanings, e.g. Q does not indicates any channel change. .

33. Line 208-210. Reference someone who showed the extent of changes -- or make a new map.

Line 208-210 describes one of our novel findings – that the annual flood extent within the Cambodian floodplains is reduced – so we are not able to oblige the reviewer's request to add a citation here.

We have not amassed sufficient information to precisely plot the exact extent of changes across the Cambodian floodplains. Such an attempt is out of the scope of the paper. Alternatively, there are remote sensing-based works that have showed the change in flood extent within the TSL, should the reviewer be interested (Ji et al., 2018; Lin and Qi, 2017; Wang et al., 2020).

34. Lines 225 -230. Provide more details and show on maps if possible.

The irrigation canals and infrastructure are shown in Figure 1.

35. Line 235. This "hint" has been demonstrated by others and they should be cited.

Line 232 to 236 contain another one of our novel findings – that rise rates and fall rates have increased within the Cambodian floodplains – so we are not able to oblige the reviewer's request to add a citation here.

Instead, we changed the word 'hint' to 'point' to reduce ambiguity. Thank you for your suggestion.

36. Line 250. Figure 4b (maybe 5b, I have two versions of this draft). Maybe make the shaded area here light blue, as it is not the same period as the light gray one in plot a. Improves readability.

Thank you for your suggestion. We have modified the graph accordingly.

37. Lines 265 to 271. This information needs to be carefully presented. How do dams that are "being developed" affect the past flows analyzed in this paper? The dam at Stung Prasat was only 80-90% finished in May 2021. Where is Stung Sreng? All these places need to be shown on the maps. This paragraph is highly speculative.

We apologise for the miscommunication. We cited Stung Sreng and Stung Pursat as examples of water infrastructure developments along the TSL's tributaries. We do not mean that these reservoirs are responsible for the hydrological changes in the lake. As such, we have edited this section to reduce ambiguity for our readers.

38. Figure 6. Schematic of wet season discharge on the Cambodian floodplains during the pre-dam and mega-dam era. Across all stations, there is a reduction of discharge during the mega-dam era of 2010-2019. This is interesting, but the black arrows "admit" to the uncertainty of the flows and the difficulty in interpreting a few stage and Q measurements. I don't think the Q values listed agree entirely with your Qdiff values from before. Do they tell the same story? Please take a look and comment if needed.

Thank you for the concern. We cross checked the values in Figure 6, 7, 8 and Table 2 and found the readings consistent.

The Q measurements in Figure 6 are not exactly commensurate with Qdiff values – they tell different aspects of the story. Figure 6 showed how the reduction of wet-season discharge (already established in Section 4.1) varies across the Cambodian floodplains. We found that the developments upstream of Stung Treng (be it Chinese or Laotian dams) cannot fully account for the reduction in flows further downstream.

The calculation of Qdiff in Figure 7 and 8 then further supports this narrative by showing that there is an increase in water lost through the floodplains. Further analysis then pinpoints the source of this loss to be local anthropogenic factors, which we argue to be water withdrawal in the region.

39. Figure 6. What does the annual flux look like (put in SI)?

Thank you for your suggestion. We will add in the unit accordingly in Figure 6. We are not considering annual flux but only wet-season flux from June to September.

40. Section 5.2. To fully understand this potential impact the reader needs to know the location of the rice fields, and they need to know more about the rainfall: the droughts of 2015 to 2018 were prolonged and extensive. Your rainfall analysis is lacking in rigorousness as it includes only two stations and two months considered. IMPORTANT LIMITATION.

The locations of the canals have already been marked on in Figure 1. We are unsure of the reviewer's claim of a prolonged drought from 2015 to 2018. We could not find any reports that refer to it. Perhaps the reviewer meant the drought from June 2019 to August 2021 afflicting the region. Nonetheless, the 2019-2021 drought is not within the study period of our study.

For our rainfall analysis, we included daily data from 1960-2019 plotted in quarterly intervals, not "two months" as the reviewer has suggested. As mentioned, our precipitation analysis is not for the purpose of budgeting or modelling, but only to illustrate the point that there has not been major changes to rainfall trends in the Cambodian floodplains from 1960-2019. Furthermore, our analysis based on ground measurements mirrors the result by other studies that used other sensing methods (Raghavan et al., 2018; Singh and Qin, 2020; Thoeun, 2015).

41. Section 5.3. Check out the Ng and Park (2021) paper, then rewrite this section. You will need to explain in better detail the processes by which sand mining below the lake is affecting flows, then relating this to observed discharges of water into the lake.

Thank you for the suggestion.

In Ng and Park (2021), the authors described sand mining at Phnom Penh, not "below the lake" as the reviewer has alleged. Thus, the link between sand mining and TSL discharge is not what the reviewer has presumed.

Furthermore, the findings of Ng and Park (2021) differ from another similar (and reputable) research paper by Hackney et al. (2021). Thus, the exact quantification of sand mining within the Cambodian floodplains is still in debate and we feel that we should not favour the results of any author without proper considerations of its methods and analyses.

42. Section 5.4. Unless you are willing to go into detail the reference to the other systems is not informative. The attempt to look at wider implications is ok, though somewhat speculative. I think it is important to compare your finding(s) with the work of others, but it seems to me they basically reinforce what is largely known. You should discuss limitations of your approach with respect to errors and limitations.

Thank you for your suggestions. We disagree with the reviewer's allegation that our findings are "largely known". For example, to our best knowledge, our quantification of the reduction of reverse flow to the TSL has not been replicated by any studies prior.

We have added in an additional Section 5.5 that discuss the limitations and direction of future studies within the Cambodian floodplains.

43. Regarding the nature of the special issue on transboundary issues, you may want to make a small section on this aspect to add value to the paper. In particular, insights regarding social aspects of this transboundary issue is greatly missing, given the expertise on the Tonle Sap in Singapore, both present and past.

Thank you for your suggestions. Indeed, we have discussed the transboundary nature of the Cambodian floodplains. In Section 5.1, we discussed the impacts of upstream dams from China or Laos on the flood pulse in the floodplains. Through our analysis, we found that the reduction of wet-season discharge in the Cambodian floodplains cannot be attributed fully to these transboundary drivers.

In Section 5.4, we further discussed the potential impact to the Vietnamese Mekong Delta, located downstream of the Cambodian floodplains. Hopefully, through our study, we hope that readers can derive a scientific and objective view of hydrological processes occurring within the floodplains, instead of opinions commonly purported by the mass media.

44. Conclusion. Make sure to highlight new findings (explain novelty), not just summarize results. I like the attempt in the last paragraph to emphasize the importance of the loss of the flood pulse, regardless of the reason(s). I am not sure what "water harvesting" means; is it in reference to irrigation?

Thank you for your suggestion. We have rewritten the section for clarity. "Harvesting" in this sense here refers to the usage of Mekong, be it for its water for irrigation, its fisheries, its sediment or for hydropower uses.

We thank the reviewer for time taken to read our manuscript.

References:

Arias, M. E., Piman, T., Lauri, H., Cochrane, T. A. and Kummu, M.: Dams on Mekong tributaries as significant contributors of hydrological alterations to the Tonle Sap Floodplain in Cambodia, Hydrol. Earth Syst. Sci., 18(12), 5303–5315, doi:10.5194/hess-18-5303-2014, 2014.

Cochrane, T. A., Arias, M. E. and Piman, T.: Historical impact of water infrastructure on water levels of the Mekong River and the Tonle Sap system, Hydrol. Earth Syst. Sci., 18(11), 4529–4541, doi:10.5194/hess-18-4529-2014, 2014.

Hackney, C. R., Vasilopoulos, G., Heng, S., Darbari, V., Walker, S. and Parsons, D. R.: Sand mining far outpaces natural supply in a large alluvial river, Earth Surf. Dyn., 9(5), 1323–1334, doi:10.5194/esurf-9-1323-2021, 2021.

Hecht, J. S., Lacombe, G., Arias, M. E., Dang, T. D. and Piman, T.: Hydropower dams of the Mekong River basin: A review of their hydrological impacts, J. Hydrol., 568, 285–300, doi:10.1016/j.jhydrol.2018.10.045, 2019.

Ji, X., Li, Y., Luo, X. and He, D.: Changes in the Lake Area of Tonle Sap: Possible Linkage to Runoff Alterations in the Lancang River?, Remote Sens., 10(6), 866, doi:10.3390/rs10060866, 2018.

Lin, Z. and Qi, J.: Hydro-dam – A nature-based solution or an ecological problem: The fate of the Tonlé Sap Lake, Environ. Res., 158(June), 24–32, doi:10.1016/j.envres.2017.05.016, 2017.

Lu, X. X. and Chua, S. D. X.: River Discharge and Water Level Changes in the Mekong River: Droughts in an Era of Mega-Dams, Hydrol. Process., 35(7), doi:10.1002/hyp.14265, 2021.

Ng, W. X. and Park, E.: Shrinking Tonlé Sap and the recent intensification of sand mining in the Cambodian Mekong River, Sci. Total Environ., 777, doi:10.1016/j.scitotenv.2021.146180, 2021.

Piman, T., Lennaerts, T. and Southalack, P.: Assessment of hydrological changes in the lower mekong basin from basin-wide development scenarios, Hydrol. Process., 27(15), 2115–2125, doi:10.1002/hyp.9764, 2013.

Pokhrel, Y., Burbano, M., Roush, J., Kang, H., Sridhar, V. and Hyndman, D.: A Review of the Integrated Effects of Changing Climate, Land Use, and Dams on Mekong River Hydrology, Water, 10(3), 266, doi:10.3390/w10030266, 2018.

Raghavan, S. V, Liu, J., Nguyen, N. S., Vu, M. T. and Liong, S.-Y.: Assessment of CMIP5 historical simulations of rainfall over Southeast Asia, Theor. Appl. Climatol., 132(3–4), 989–1002, doi:10.1007/s00704-017-2111-z, 2018.

Singh, V. and Qin, X.: Study of rainfall variabilities in Southeast Asia using long-term gridded rainfall and its substantiation through global climate indices, J. Hydrol., 585, 124320, doi:10.1016/j.jhydrol.2019.124320, 2020.

Thoeun, H. C.: Observed and projected changes in temperature and rainfall in Cambodia, Weather Clim. Extrem., 7, 61–71, doi:10.1016/j.wace.2015.02.001, 2015.

Wang, Y., Feng, L., Liu, J., Hou, X. and Chen, D.: Changes of inundation area and water turbidity of Tonle Sap Lake: responses to climate changes or upstream dam construction?, Environ. Res. Lett., 15(9), 0940a1, doi:10.1088/1748-9326/abac79, 2020.